# Enhanced future changes in wet and dry extremes over Africa at convection-permitting scale

Elizabeth J. Kendon[1], Rachel A. Stratton[1], Simon Tucker[1], John H. Marsham[2], Ségolène Berthou[1], David P. Rowell [1] & Catherine A. Senior[1]

African society is particularly vulnerable to climate change. The representation of convection in climate models has so far restricted our ability to accurately simulate African weather extremes, limiting climate change predictions. Here we show results from climate change experiments with a convection-permitting (4.5 km grid-spacing) model, for the first time over an Africa-wide domain (CP4A). The model realistically captures hourly rainfall characteristics, unlike coarser resolution models. CP4A shows greater future increases in extreme 3-hourly precipitation compared to a convection-parameterised 25 km model (R25). CP4A also shows future increases in dry spell length during the wet season over western and central Africa, weaker or not apparent in R25. These differences relate to the more realistic representation of convection in CP4A, and its response to increasing atmospheric moisture and stability. We conclude that, with the more accurate representation of convection, projected changes in both wet and dry extremes over Africa may be more severe.

[1] Met Office Hadley Centre, Fitzroy Road, Exeter EX1 3PB, UK. [2] Institute for Climate and Atmospheric Science, University of Leeds, Leeds LS2 9JT, UK. Correspondence and requests for materials should be addressed to E.J.K. (email: elizabeth.kendon@metoffice.gov.uk)

Until now, future climate projections across Africa have been provided by relatively coarse resolution (order 10–100 -km grid spacing) climate models[1,2]. These models rely on a parameterisation scheme to represent the average effects of convection; this simplification is a known source of model error, especially in the tropics where convection is integral to circulation and extremes. Convection parameterisation schemes often produce very intermittent rainfall at the model time step[3], rainfall too early in the day[4], and fail to capture organised propagating systems, instead simulating unrealistically widespread light daily total and insufficient heavy rain[5]. Such deficiencies can have a significant impact on the regional scale circulation and water cycle over Africa[6,7] and the response of storms to their environment[8]. For Africa, in common with much of the tropics, there is currently significant uncertainty in climate-change projections, with disparity in the sign of rainfall change in some regions[1,2,9–12]. This uncertainty makes it difficult to predict the impacts of climate change and develop adaptation strategies.

Models with order 1 -km grid spacing can represent convection explicitly without the need for parameterisation. Such models are termed 'convection-permitting' because larger storms and mesoscale convective organisation are permitted, even if smaller storms are not well resolved. Convection-permitting models (CPMs) are able to better simulate the diurnal cycle of tropical convection[7], the vertical cloud structure and the coupling between moist convection and convergence[13,14] and soil moisture-convection feedbacks in the Sahel[15]. The benefits of CPMs have also been demonstrated in other regions, including a more realistic representation of the precipitation structure and extremes[16]. The improved realism of CPMs is a key indicator of their skill in representing the underlying processes, and hence our confidence in their projections of future change. CPMs therefore present a crucial opportunity for Africa, providing insight into potential biases that exist across global model projections due to the necessary use of parameterised convection.

CPMs are used routinely for weather forecasting[17,18], but their use in climate studies has largely been limited to small domains or single seasons due to high computational cost. Long convection-permitting climate change runs have almost entirely been focussed over developed mid-latitude regions[16], with no such runs for Africa.

Here, for the first time we analyse multi-year climate-change projections for an Africa-wide domain at convection-permitting (4.5 km) resolution (CP4A). This builds on an earlier study[19] that described the CP4A experimental design and provided the results from the first 5 years of the present-day simulation. We compare future changes in 3-hourly rainfall across Africa in CP4A with the results from a 25 -km regional climate model (R25). Both models are run for 10-year present-day (1997–2007) and 10-year future (~ 2100, under Intergovernmental Panel on Climate Change RCP8.5 scenario) periods, driven by a 25 -km global model. R25 has similar model physics to CP4A (except that convection is parameterised, and it uses a different cloud and boundary layer scheme) and it was run to isolate the impact of convection parameterisation and resolution on African climate variability and change. Present-day rainfall has been compared with satellite-derived 3-hourly rainfall available from TRMM[20] and bias-corrected CMORPH[21] for 1998–2008. We focus on the wettest season, as the 3-month period with the highest mean precipitation at each location.

We find that CP4A shows greater future increases in extreme 3-hourly precipitation compared with R25. The scaling of these changes with increased atmospheric moisture is higher in the convection-permitting model, which may be explained by local dynamical feedbacks within storms amplifying increases in rainfall extremes on hourly timescales. CP4A also shows future increases in dry spell length during the wet season over western and central Africa, weaker or not apparent in R25. These dry spell changes appear to be related to the more realistic triggering and propagation of convection in CP4A and their response to future increases in stability. We conclude that changes in extreme rainfall and dry spells over Africa may be underestimated in all models where convection is parameterised.

## Results

**Representation of wet season precipitation.** The wettest 3-month period varies across tropical and sub-tropical Africa: from boreal summer in the north to boreal winter in the south, with the equatorial regions dominated by one or other of the periods when the Intertropical Convergence Zone (ITCZ) crosses the equator (Fig. 1). R25, CP4A and CMORPH all show a similar spatial variation in the wet season to TRMM, although there are discrepancies over northern Africa and neither model captures the variation between March–April–May (MAM) and October–November–December (OND) across East Africa. CP4A better captures the wet season over the West African monsoon region (1 month too early in R25). Some future changes in wet season are seen in both models: over West Africa, the wet season is typically shifted 1-month later, consistent with previous studies[11]; over southern Africa, it is shifted 1-month earlier and over the Horn of East Africa, there is a shift from MAM to OND as being the wettest period in the future, shown to relate to a slower retreat of the ITCZ southwards[22]. For consistency, in all following analysis, we use the present-day wet season in TRMM as a common definition for all datasets, but note this could lead to an underestimation of future increases in heavy precipitation.

CP4A gives a better representation of 3-hourly precipitation in the wet season across Africa than R25 (Figs. 2–6). Rainfall in R25 (almost entirely from the convection scheme) is too frequent and too light, with heavy precipitation being considerably too low. Although CP4A has similar biases in the mean, over land it gives a much better representation of rainfall occurrence, intensity and extremes. In CP4A, wet season rainfall is less frequent but more intense over most of tropical Africa, confirming previous RCM studies that showed a similar result on increasing resolution[9]. CP4A tends to overestimate rainfall over ocean along the ITCZ, which may be caused by the uncoupled ocean[23], and both R25 and CP4A have a tendency to overestimate rainfall over mountains and lakes, although this is more pronounced in CP4A.

**Future change in extreme precipitation.** In the future, both models show increases in mean and extreme precipitation across much of Africa (Figs. 2, 6). Decreases in mean precipitation are seen in some west and south–west regions, with decreases in rainfall occurrence; in these regions, there is still a tendency for rainfall intensity and extremes to increase. These results, in terms of the pattern of change and increases in the ratio of extreme to mean rainfall, agree qualitatively with previous studies[10,24,25]. Compared with R25, CP4A shows slightly smaller increases in mean precipitation; but importantly, CP4A shows much larger decreases in rainfall occurrence (18% compared with 1.5% in R25) and larger increases in precipitation intensity (39% compared with 22%) and extremes (Figs. 3–6). CP4A's smaller increase in mean precipitation over the Congo, corresponds to its greater reduction in rainfall frequency there. These differences between CP4A and R25 show how changing convection affects much larger scales. In terms of extremes, exceedance of the present-day 99.9th-percentile occurs almost three times more frequently in the future compared with the present-day across Africa in CP4A (Fig. 6).

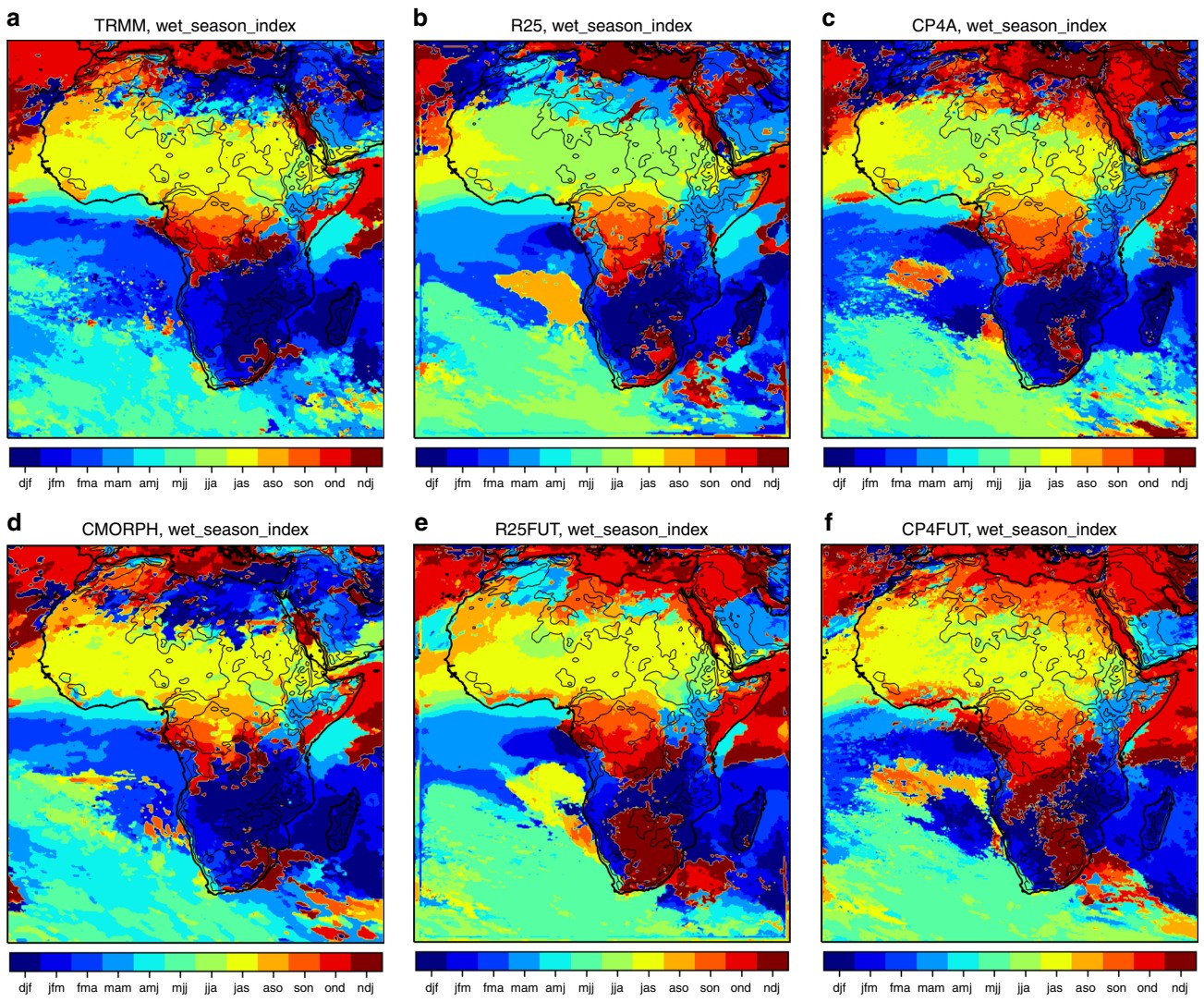

**Fig. 1** Wet season index. Three-month period with the highest mean precipitation in **a**, **d** TRMM, CMORPH and **b**, **e** R25 model, for present day and future, and **c**, **f** CP4A, for present day and future. The black lines indicate the 500 -, 1000-, 2000-, 3000- and 4000 -m height contours. For Figs. 2–6, 10 and Supplementary Figures 11–16, wet season as observed in TRMM is used as a common definition for all datasets

To explore future changes in the intensity distribution, we consider the fractional contribution of different 3-hourly intensity bins to the total precipitation, for sub-regions (Fig. 6) sampling different climates across Africa. This demonstrates the improved performance of CP4A for the present-day (shown for Sahel and E-Africa in Fig. 7 and for other regions in Supplementary Figs. 1, 2). TRMM tends to miss low-intensity events and have more high-intensity events compared with CMORPH; this difference is more pronounced with the new bias-corrected CMORPH as opposed to the original CMORPH-v1. However, CP4A gives better agreement with either observational dataset than R25 (except perhaps compared with CMORPH for the Gulf-of-Guinea and S-Africa, but note there is an odd feature in CMORPH near Guinea, see the Methods section). In the future, CP4A consistently shows greater increases in the fractional contribution from high rain rates compared with R25, mainly because of more high rain rates in absolute terms (greater increases are also seen for an equivalent high percentile, Fig. 6). In CP4A, exceeding 60 - mm accumulation in 3 h, at the 25 -km scale, is 7–8 times more frequent in future compared with the present-day for the Sahel and E-Africa (and 4–6 times more frequent in other regions). In these regions, such an event shifts from occurring typically once

in every 30 years at each grid point in the present-climate to once every 3–4 years in the future. In R25, such exceedances are rarer and typically increase less in future; over East Africa, the future increase (of 9×) is similar, but such events are 8× rarer than those in CP4A. These relatively rare events may be flood-inducing. The highest ever recorded event in Dakar in August 2012, with 144 mm in 1 h (161 mm in 6 h) at the local station scale, led to widespread flooding with 287,000 people displaced and 18 deaths[26]. At the 25 -km scale, the rainfall total is expected to be much less than at the point scale (an areal reduction factor of 0.82 is estimated between point and 25 -km rainfall data in the Sahel[27], but the factor is likely to be lower for localised extreme events). Overall, 60 mm in 3 h is lower than the Dakar flood-producing event, but such a high accumulation over the 25- km scale may lead to local flash flooding in some cases[28], and is chosen as a user-relevant threshold for which we can estimate more robust statistics of future change.

**Future change in dry spells**. We see notable differences in future changes in the length of dry spells between CP4A and R25 (Fig. 8). For a 1 mm day$^{-1}$ threshold, R25 underestimates the

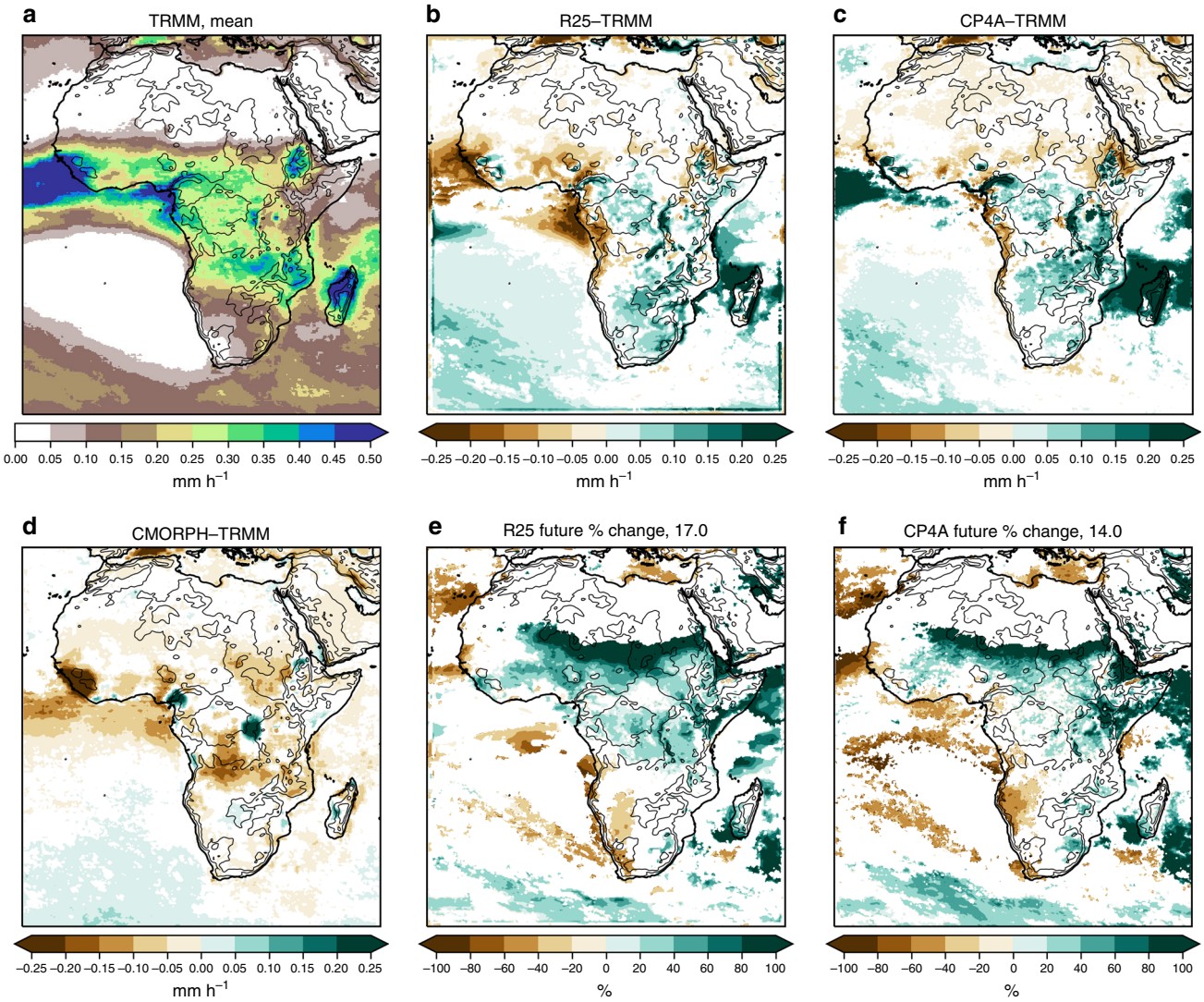

**Fig. 2** Wet season mean precipitation. **a** TRMM observations, differences with respect to TRMM for **b** the R25 model, **c** CP4A model and **d** CMORPH observations, and percentage differences between 2100 and present day for **e** the R25 model and **f** CP4A model. The median of future percentage changes across Africa (land points only) is indicated in **e**, **f**. Dataset differences and future changes are masked in white, where differences are not significant at the 5% level compared with year-to-year variability. The wet season is the 3-month period with the highest mean precipitation in TRMM, defined on a grid-point basis. The black lines indicate the 500 -, 1000 -, 2000-, 3000 - and 4000 -m height contours

length of dry spells in two of the five regions during their respective wet seasons, and overestimates them in Central Africa (Supplementary Figs. 3, 4). In each case, CP4A reduces this bias. In the future, CP4A shows a significant lengthening of dry spells during the wet season over the Sahel, Gulf-of-Guinea and Central Africa; over the Gulf-of-Guinea, dry spells exceeding 10 days are almost twice as frequent in the future compared with the present-day. This lengthening of dry spells is not seen (or much smaller) in R25, and over the Sahel R25 shows the reverse effect. Using a present-day percentile of the daily distribution to define dry spells gives very similar results in terms of future changes, although in this case R25 tends to overestimate the length of dry spells (Supplementary Fig. 5). Thus the model differences in dry spell changes are not simply a reflection of the different number of dry days in the present-day.

Hovmöller plots for the latitude band 5–15°N across central/western Africa show that CP4A gives a much more realistic representation of westward propagating features and the diurnal cycle than R25 (Fig. 9). In R25, the diurnal cycle is too strong and

there are some anomalous eastward moving features[19]. There are some eastward moving features in TRMM, but these are less prominent than in R25. In future, longer periods of no rain are clearly visible in CP4A (Fig. 9). These appear to result from much less diurnal triggering of rainfall (disappearance of evening storms) and also the dying out of westward propagating features, especially west of 10°E. These changes are consistent with a future increase in convective inhibition, with a decrease in the number of profiles able to support convection from the surface layer during the day (Supplementary Figs. 6, 7). Similar results are seen in CPM simulations over the United States[29]. R25 shows a weaker effect: the diurnal cycle remains strong in the future, along with propagating features west of 10°E (Supplementary Figs. 8–10). Thus the difference in dry spell changes between the models appears to be due to the improved representation of diurnal convection and propagating systems in CP4A, and their response to increasing stability. We note, runoff does not increase more in CP4A (Supplementary Fig. 11), so a reduction in local moisture recycling due to high future intensities (proposed by previous

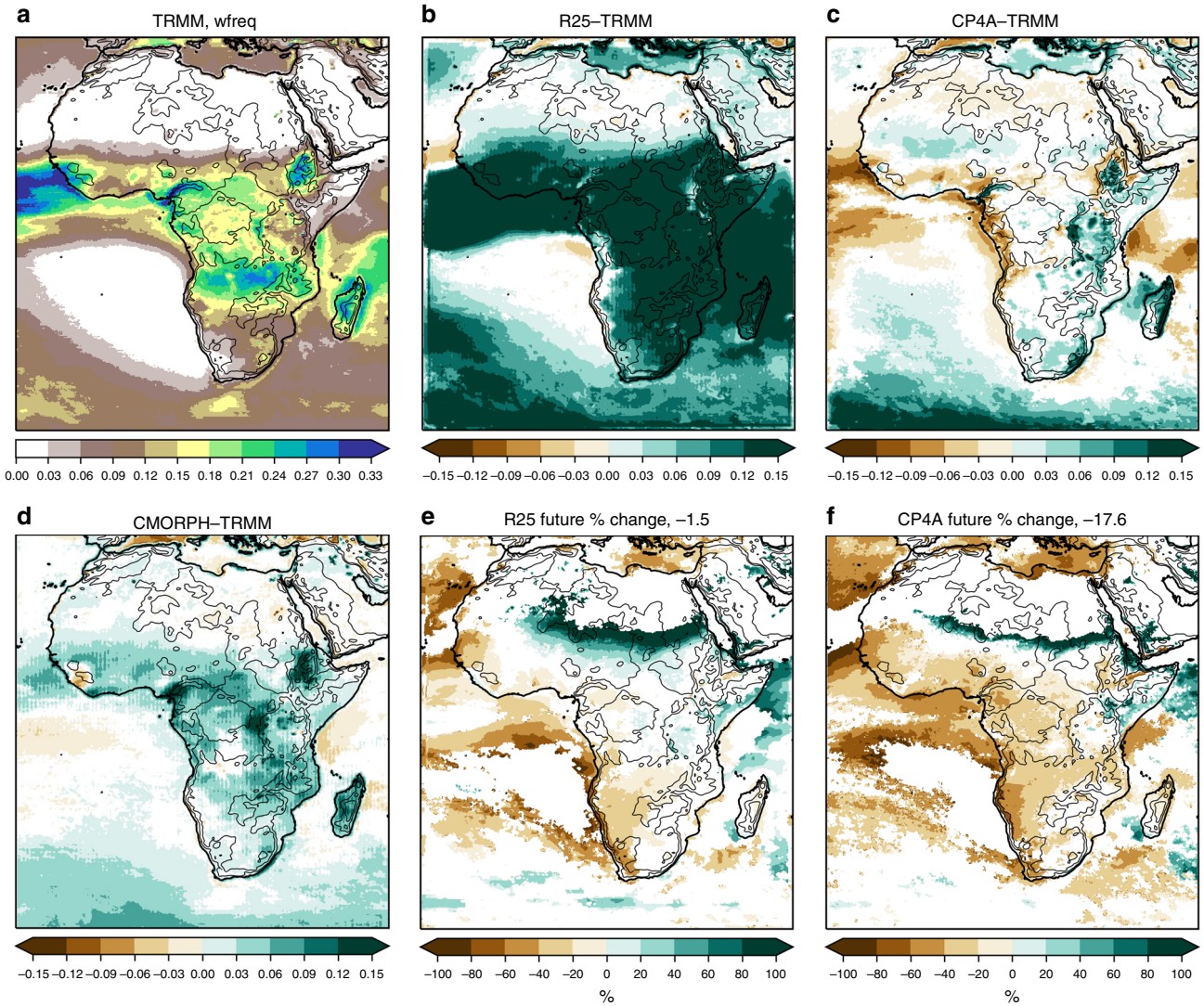

**Fig. 3** Wet season 3-hourly precipitation occurrence. **a** TRMM observations, differences with respect to TRMM for **b** the R25 model, **c** CP4A model and **d** CMORPH observations, and percentage differences between 2100 and present day for **e** the R25 model and **f** CP4A model. Precipitation occurrence is defined as the frequency of wet values (>0.1 mm h$^{-1}$). The median of future percentage changes across Africa (land points only) is indicated in **e**, **f**. Dataset differences and future changes are masked in white, where differences are not significant at the 5% level compared with year-to-year variability. The wet season is the 3-month period with the highest mean precipitation in TRMM, defined on a grid-point basis. The black lines indicate the 500 -, 1000 -, 2000-, 3000 - and 4000 -m height contours

studies of 50 -km RCMs over Africa[9] and convection-permitting simulations of the Indian monsoon[30]) does not explain the model differences in dry spell changes here.

**Scaling of extreme precipitation change.** Finally, we explore how future increases in extreme precipitation intensity relate to increased atmospheric moisture with warming across Africa, and whether this differs at convection-permitting scale. Saturated water vapour pressure increases with temperature at 6–7%K$^{-1}$, following the Clausius–Clapeyron (CC) relation. However, changes in moisture availability may be much less than this temperature-dependent maximum, and so here we consider changes in near-surface dew point temperature, which is a measure of specific humidity translated to temperature using the CC relationship[31]. We find that increases in extreme 3-hourly precipitation intensity during the wet season are typically higher than CC scaling (7.8%K.) in CP4A, but lower (5.1%K$^{-1}$) in R25 (Fig. 10; Supplementary Fig. 12 for daily extremes). In both

models, the scaling rate increases for higher percentiles of the precipitation distribution (Supplementary Fig. 13), consistent with previous studies[32], with the scaling rate consistently higher in CP4A than R25. Using a present-day percentile to define wet values, or a metric of all values, shows that the higher scaling in CP4A is not simply due to differences in the number of dry values (<0.1 mm h$^{-1}$) in the present-day (Supplementary Figs. 14, 15). When using a percentile of all values, the scaling difference between the models is smaller. A future reduction in the occurrence of low-intensity events alone (e.g., due to increasing convective inhibition) could lead to higher apparent scaling for wet-value percentiles. However, Supplementary Fig. 15 shows that the higher scaling in CP4A is not simply due to the greater increase in dry spells. Since it is the intensity and less clearly the frequency of precipitation that is directly related to increased atmospheric moisture with warming[33], we focus on extreme precipitation intensity for the scaling analysis.

We see large departures from CC-scaling locally, with evidence of 'super-CC scaling' occurring more widely across Africa in

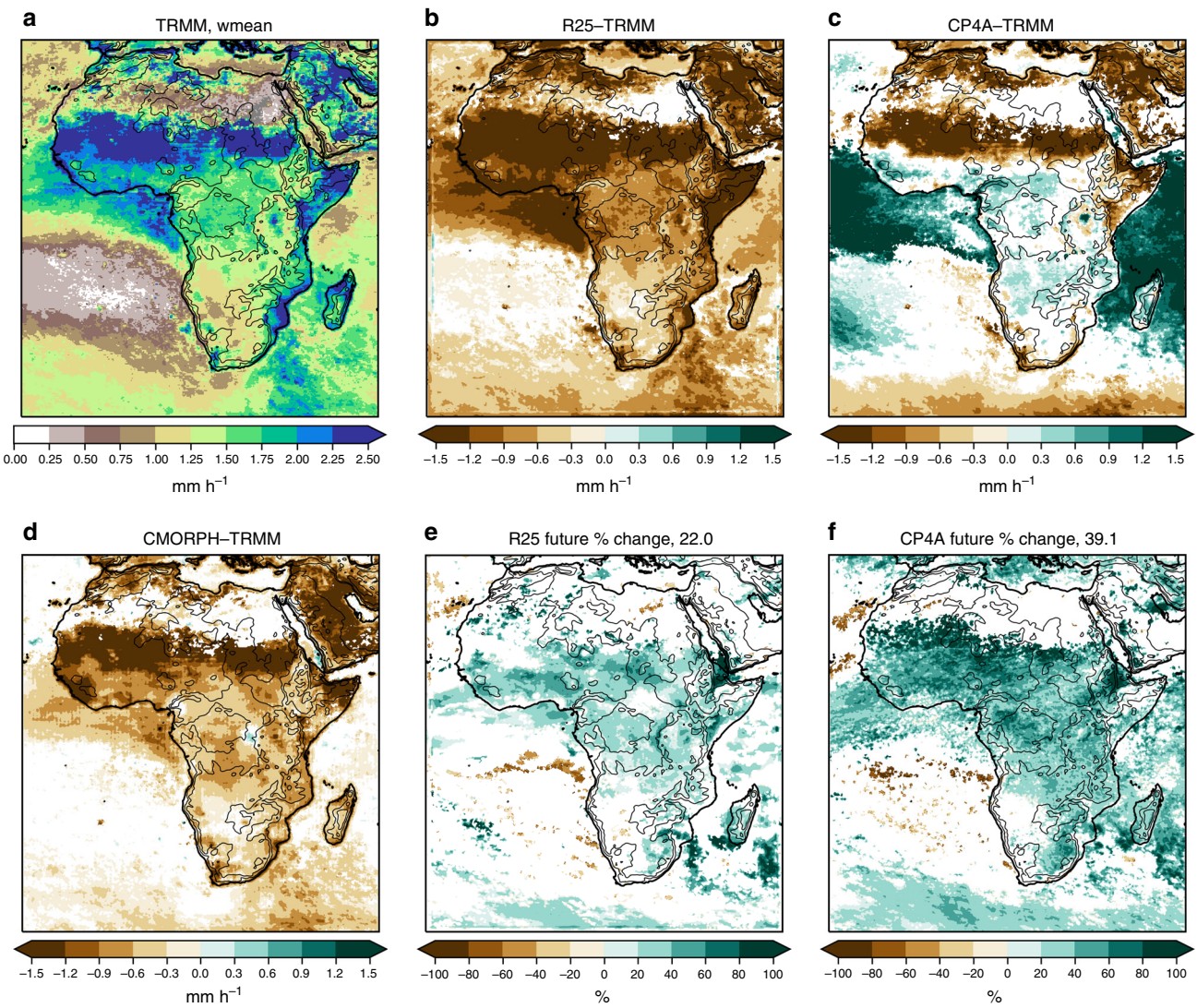

**Fig. 4** Wet season mean 3-hourly precipitation intensity. **a** TRMM observations, differences with respect to TRMM for **b** the R25 model, **c** CP4A model and **d** CMORPH observations, and percentage differences between 2100 and present day for **e** the R25 model and **f** CP4A model. Mean precipitation intensity is defined as the mean of wet values (>0.1 mm h⁻¹). The median of future percentage changes across Africa (land points only) is indicated in **e**, **f**. Dataset differences and future changes are masked in white, where differences are not significant at the 5% level compared to year-to-year variability. The wet season is the 3-month period with the highest mean precipitation in TRMM, defined on a grid-point basis. The black lines indicate the 500 -, 1000 -, 2000-, 3000 - and 4000 -m height contours

CP4A (Fig. 10; Supplementary Fig. 13). In particular, 11% of 25 - km grid points show scaling coefficients >2xCC in CP4A compared with 4% in R25. Both models also sample negative scaling rates, with 5% of grid points showing negative scaling in CP4A and 13% in R25. Locally varying negative to strongly positive scaling rates typically happen due to displacements of extreme convective storms in future, and may be symptomatic of an undersampling of extreme storms in the 10-year model simulations. The fact that super-CC scaling is more prevalent than negative scaling in CP4A (but not R25) shows that this is not simply explained by displacements of extreme storms. We note that over much of central Africa where super-CC scaling is observed in CP4A, future changes in extreme precipitation intensity (Fig. 5) are significantly higher in CP4A than in R25 taking into account year-to-year variability. These results suggest that higher scaling rates at convection-permitting scale for Africa as a whole are robust, but given the results are based on single 10-year model realisations, there is uncertainty in the actual scaling values at the 25 -km grid point scale.

On using surface temperature (instead of dew point) as the scaling variable (Supplementary Fig. 16), we see a downturn in scaling at a high-temperature change explained by moisture availability not increasing as fast as temperature, with a decrease in relative humidity. We see some consistency between models in regions showing sub-CC scaling; in particular, both models show a large area of sub-CC scaling with surface temperature change in SW Africa, consistent with regional drying. However, in general, variation in the scaling coefficient across grid points in CP4A is not strongly related to that in R25, with a correlation of 0.4 or less. At high temperature change (>8 K), scaling coefficients in the two models are uncorrelated, suggesting a different response in dry environments where future warming is likely largest.

## Discussion

The results here show that increases in sub-daily (and to a lesser extent daily) precipitation extremes are more severe in CP4A compared with a coarser resolution model, and this relates to higher scaling rates with atmospheric moisture in the

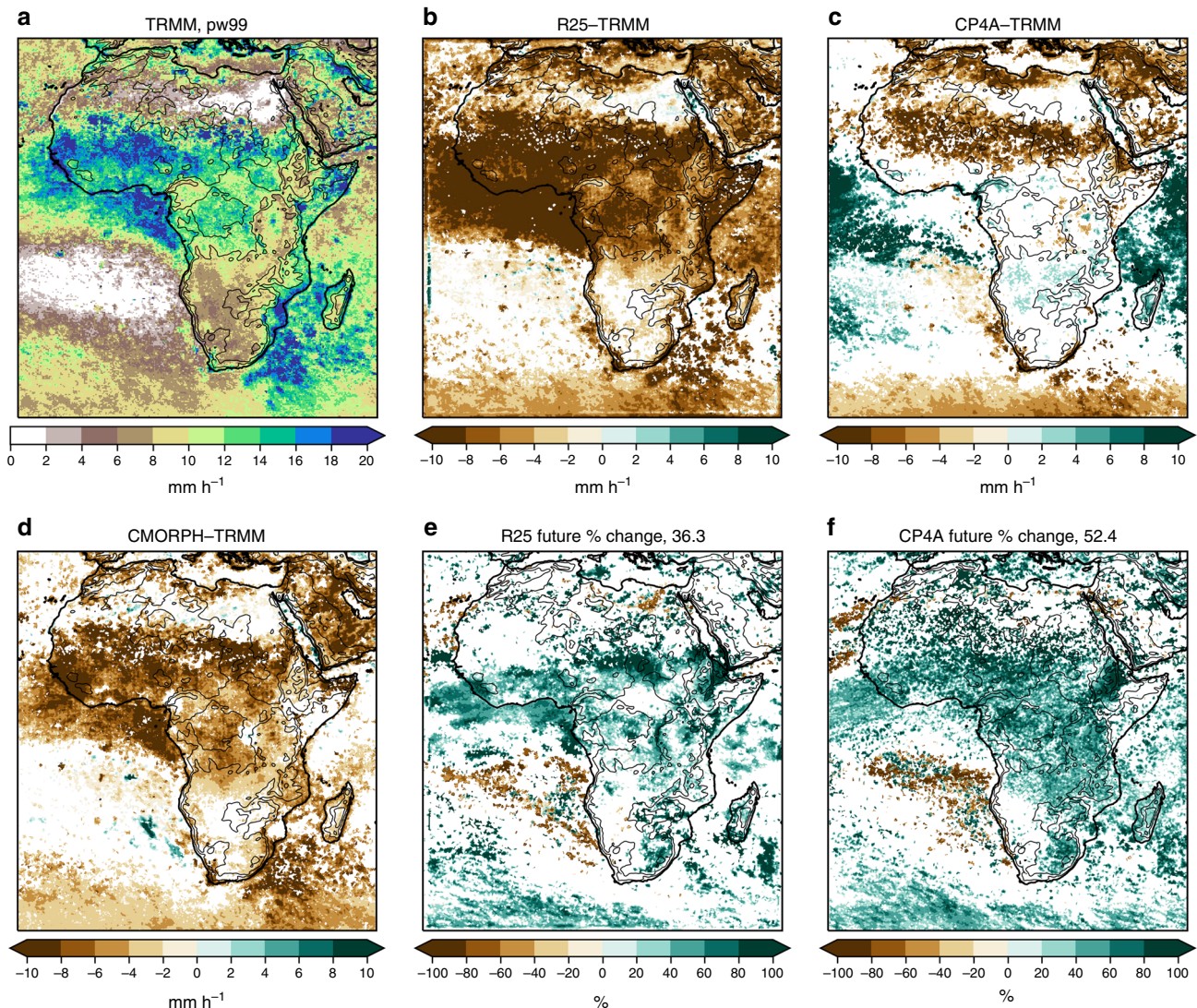

**Fig. 5** Wet season extreme precipitation intensity. **a** TRMM observations, differences with respect to TRMM for **b** the R25 model, **c** CP4A model and **d** CMORPH observations, and percentage differences between 2100 and present day for **e** the R25 model and **f** CP4A model. Extreme precipitation intensity is defined as the 99th percentile of wet values (>0.1 mm h$^{-1}$), for 3-hourly precipitation. The median of future percentage changes across Africa (land points only) is indicated in **e**, **f**. Dataset differences and future changes are masked in white, where differences are not significant at the 5% level compared with year-to-year variability. The wet season is the 3-month period with the highest mean precipitation in TRMM, defined on a grid-point basis. The black lines indicate the 500 -, 1000 -, 2000-, 3000 - and 4000 -m height contours

convection-permitting model. Departures from CC scaling with the dew point temperature may be due to a number of possible explanations. Firstly, changes in the dew point temperature at higher levels may be more representative of the changes in moisture feeding into storms (rather than changes at the 1.5 -m level used here). Secondly, the dew point temperature change for the air actually feeding into storms may be different from the mean wet season dew point temperature change used here. Thirdly, there may be changes in storm vertical velocities. Fourthly, there may be local dynamical feedbacks within storms that change moisture inflow and overall precipitation efficiency. Finally, there may be changes in the atmospheric environment, in which storms develop impacting precipitation efficiency and the scale and organisation of convection (for example, increased wind shear favours the mesoscale organisation of convection and this has led to a recent intensification of extreme Sahel storms[8]). We expect both CP4A and R25 models to be similarly affected by the first two points above. Local dynamics and feedbacks within storms and the organisation of convection (which relate to the

last three points) are not well captured by coarse resolution models, and the better representation of these processes in CP4A may explain the higher scaling in this model compared with R25. We note, local dynamical feedbacks within storms linked to latent heat release are expected to amplify increases in rainfall extremes on hourly timescales, and so may explain higher scaling for 3-hourly compared with daily precipitation extremes in CP4A (c.f. fig. 10 compared with Supplementary Fig. 12).

Recent studies assessing historic changes in observed precipitation have shown that natural variability plays a major role in parts of Africa[34]; although recent increases in rainfall over the Sahel have been linked to increased greenhouse gases[35]. The results here are based on single 10-year model realisations of climate change, nevertheless, we are able to detect clear changes in extreme precipitation at the end of the century above year-to-year natural variability. This is due to the fact that the signal of change is much larger by 2100, with changes in extreme precipitation largely explicable in terms of increasing atmospheric moisture with warming. There is no influence of oceanic decadal

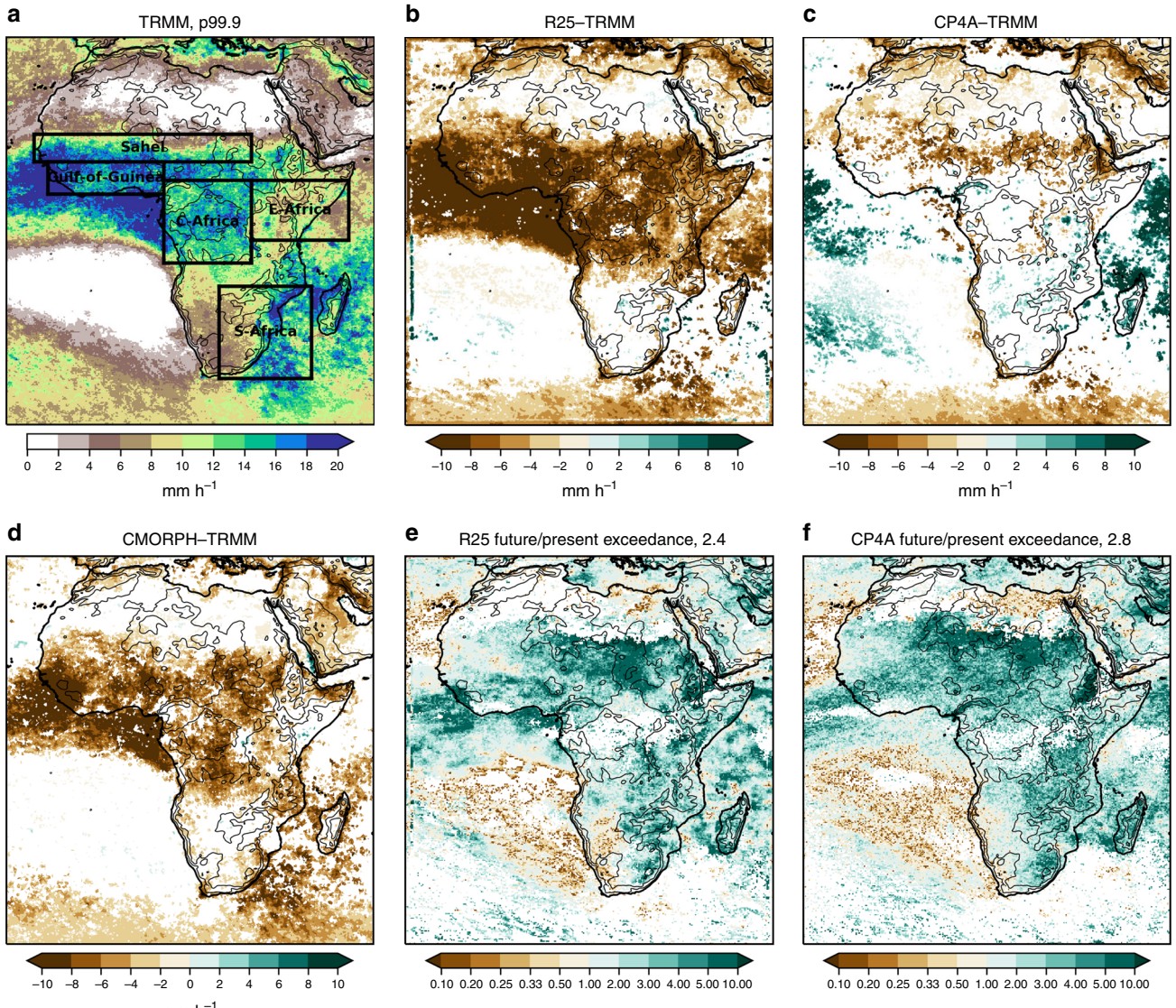

**Fig. 6** Present-day extreme precipitation and the frequency of exceedance in future, for the wet season. Present-day extreme precipitation threshold in **a** TRMM observations, differences with respect to TRMM for **b** the R25 model, **c** CP4A model and **d** CMORPH observations and the ratio of the future compared with the present-day frequency of exceedance of this threshold for **e** the R25 model and **f** CP4A model. Extreme precipitation threshold is defined as the 99.9th percentile of 3-hourly precipitation in the wet season in the present-day. The median of future/present exceedance ratio across Africa (land points only) is indicated in **e** and **f**. Differences and future changes are masked in white, where differences are not significant at the 5% level compared with year-to-year variability. For all datasets, the wet season is the 3-month period with the highest mean precipitation in TRMM, defined on a grid-point basis. The black lines indicate the 500 -, 1000 -, 2000-, 3000 - and 4000 -m height contours. Definition of Africa sub-regions for subsequent analysis is shown in **a**

variability on the results due to our adopted treatment of sea-surface temperatures (see the Methods section). The relatively short simulations here, however, do limit our sampling of extreme events and hence our ability to robustly characterise changes in the extreme tail of the precipitation distribution at the local scale.

This study indicates the importance of the representation of local convective processes for predicting future changes in precipitation extremes across Africa. This is the case not only for wet extremes where local dynamics and feedbacks within storms, which may explain departures from CC scaling and are not captured by coarse resolution models, may be important in explaining enhanced increases in CP4A compared with R25, but also for dry extremes. In particular, we see a greater tendency for increases in dry spell length in the CPM, related to the more

realistic triggering and propagation of convection. These results are for a single model, and so it is not possible to estimate modelling uncertainty. However, consistency with CPM climate projections for other regions, in particular with regard to the greater intensification of hourly rainfall with warming compared with conventional climate models[36,37], gives us greater confidence in these results. The lack of a strong relationship between temperature-precipitation scalings in CP4A and R25 suggests that enhanced ('super-CC') increases in extreme precipitation in the CPM cannot easily be diagnosed from parameterised models, and thus more CPM simulations for Africa are needed to account for uncertainty. As these become available, it is hoped they may offer a more consistent picture of future change, reducing uncertainty in the impacts of climate change across a uniquely vulnerable region.

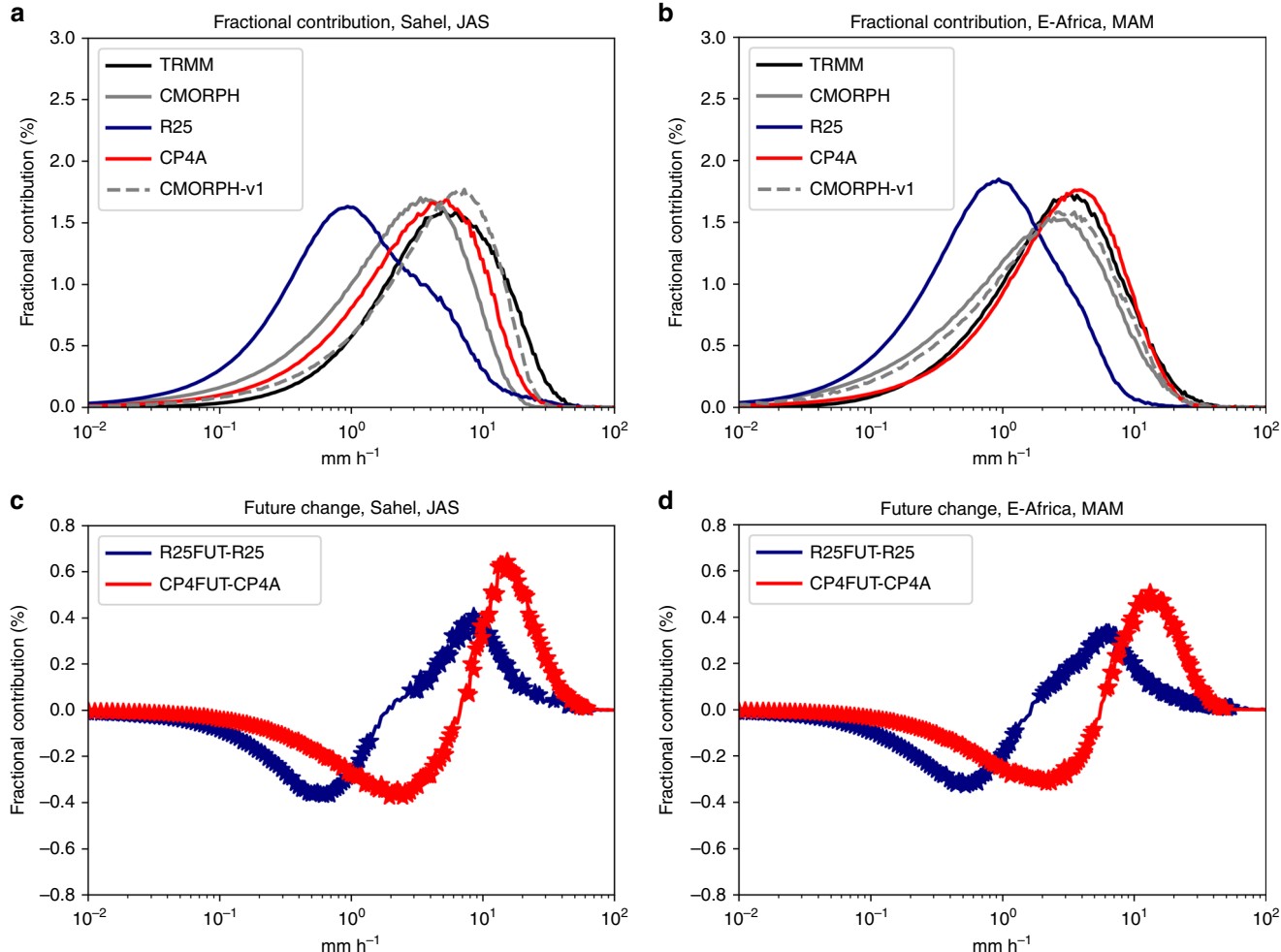

**Fig. 7** Fractional contribution of 3-hourly precipitation intensity bins to the total precipitation. **a**, **b** Fractional contribution (%) for present day for TRMM and CMORPH observations and R25 and CP4A models; **c**, **d** difference in fractional contribution between 2100 and present day for R25 and CP4A models for Sahel July–August–September (JAS) and E-Africa March–April–May (MAM). For CMORPH, the original version 1 of the data (CMORPH-v1, grey dashed) is shown as well as the bias-corrected data (CMORPH, grey solid). Regions are as shown in Fig. 6. All 3-hourly data in the given season, in the 10-year period, from all land points in the sub-region are used to calculate the fractional contribution. Stars indicate where future changes are significant at the 1% level compared with year-to-year variability, assessed using bootstrap resampling

## Methods

**Models**. The models used here are all configurations of the Met Office Unified Model (UM). The 4.5 -km CP4A model spans the African continent, and is driven by a N512 resolution global climate model (GCM). It is based on the UKV Met Office regional model that has been in use for operational numerical weather prediction since 2012. The model physics and dynamics used in CP4A have been described previously[19]. The 25 -km regional model (R25) and global model, both have the same physics configuration: a prototype version of the UM Global Atmosphere 7.0 (GA7) configuration[38] (GA7 is the atmospheric component of the Global Coupled Model 3.0 which is the UK's submission to CMIP6[39]). R25 has the same domain, land surface and aerosol forcing as CP4A and is similarly forced by the N512 GCM at its lateral boundaries. The key physics differences between the CP4A and R25 models are that in CP4A the convective parameterisation scheme is switched off, a different cloud scheme and a blended boundary layer scheme are used, and moisture conservation is applied[19].

Soil moisture evolves freely using the JULES land surface scheme[38]. CP4A and R25 use the same land-surface properties, and in particular the soil properties are defined to be spatially uniform (and those of sand) across the whole domain. This differs from the driving GCM that uses the Harmonized World Soil Database, which is considered to contain unrealistic small-scale variability across Africa[40]. In both CP4A and R25, the soil moisture fields were initialised with data derived from an off-line JULES simulation[19].

In the present-day simulations, all models are forced with sea surface temperatures (SSTs) from the Reynolds daily observations[41]. Lakes are treated differently in the two regional models compared with the N512 global model. In the global model, only Lake Victoria is modelled as a 'sea point' in the land sea mask; all other lakes are land points and treated as inland water by JULES, which

models them using an adaptation of the vegetation canopy code. The regional models treat lakes as sea points, and hence it is necessary to provide them with a surface temperature. For the 89 inland lakes that are included in the ARC-Lake v3 dataset[42], a climatology of monthly night-time lake surface temperatures (LSTs) from this dataset is used in both the CP4A and R25 models. For other inland lakes (typically those with a surface area of less the 50 km²), a surface temperature value from the model's nearest sea point is assumed. For ozone, a monthly climatology from the SPARC-II dataset[43] is used, whilst monthly climatological aerosols are derived from a GA6 simulation with interactive aerosols[44]. Greenhouse gas mass mixing ratios are varied annually, with carbon dioxide varying from $5.51679 \times 10^{-4}$ kg kg$^{-1}$ for 1997 to $5.81488 \times 10^{-4}$ kg kg$^{-1}$ for 2006. The present-day simulations span a 10-year period March 1997–February 2007; the first 2 months (January–February 1997) were discarded to allow for model spin up.

The future simulations correspond to a 10-year period around 2100, for the IPCC Representative Concentration Pathway (RCP) 8.5 climate-change scenario. Future forcings are specified following a similar approach to that of the UPSCALE project[45]. Namely, the SSTs are the sum of the SSTs used in the present-day simulations and the climatological average SST change between 1975–2005 and 2085–2115 in a HadGEM2-ES RCP8.5 run[46,47]. These SST changes were calculated for each calendar month, interpolated in both space and time, and added to the daily varying Reynolds forcing data on the various model grids. The increase in SST forcing equates to a global mean SST increase of just under 4 K[45], giving a global mean 1.5 m air temperature change of 5.2 K for the period of the CP4A simulations. For the N512 GCM, consideration also needed to be given to changes in sea ice. Greenhouse gas values were taken from the RCP8.5 climate-change scenario for the year 2100. The same ozone and aerosol climatologies are used in both the future and present-day simulations.

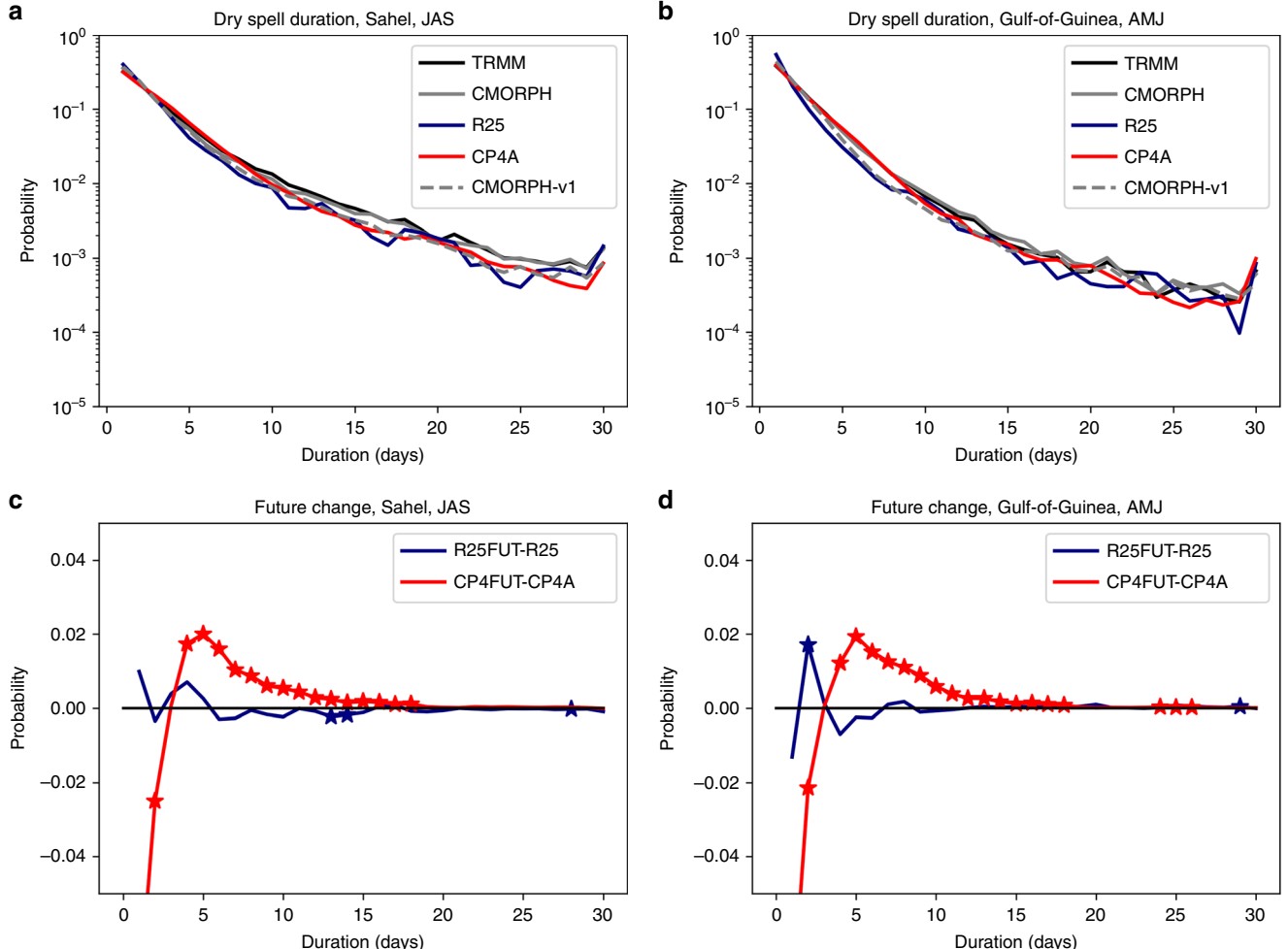

**Fig. 8** Probability distribution of dry spell duration. **a**, **b** Present-day duration of dry spells for TRMM and CMORPH observations and R25 and CP4A models; **c**, **d** differences in dry spell distribution between 2100 and present day for R25 and CP4A models for Sahel July–August–September (JAS) and Gulf-of-Guinea April–May–June (AMJ). Dry spells are defined as days with <1 mm of rainfall. For CMORPH, the original version 1 of the data (CMORPH-v1, grey dashed) is shown as well as the bias-corrected data (CMORPH, grey solid). Regions are as shown in Fig. 6. All daily data in the given season, in the 10-year period, from all land points in the sub-region are used to calculate the distribution. Stars indicate where future changes are significant at the 1% level compared with year-to-year variability, assessed using bootstrap resampling

Lake surface forcing for the future CP4A simulation is computed as the sum of the ARC-Lake observations and a seasonally varying change in LST specified from the N512 GCM 1997–2007 and the corresponding time period in the future simulation monthly climatologies. All African lakes in N512 GCM, except for Lake Victoria, are land points. We therefore choose to model the change in LST as the sum of the N512 GCM change in local land surface temperature (75% weighting, using all GCM grid boxes at least partly overlapping the lake) and an inertia from the continental coastal land–sea temperature contrast (25% weighting). The inertial term is given low weight due to recent evidence that many lakes may be currently warming as fast as neighbouring land[48], although this is yet to be substantiated by other studies. To avoid double counting of lake effects by accidentally including them in the local land surface temperature changes, all grid boxes containing inland water and lake Victoria were masked in the N512 GCM simulation, and replaced with temperature values specified as the average of a five grid box ring of 100% land boxes surrounding each lake (a lake being connected masked grid boxes). The inertial term is computed as the ratio of coastal ocean-to-land warming, using a band of five N512 GCM grid boxes either side of the coastline surrounding Africa from latitudes 15°S–15°N. The change in LST is computed separately for each lake and for each month, with 3-month smoothing applied to its annual cycle. Only Lake Victoria is present in the N512 GCM, where its change in LST is founded on HadGEM2-ES N96 data using the same weighted-sum approach as above (but with the band of grid boxes used to calculate the coastal land–sea temperature contrast being just one grid box thick). Finally, for all R25 lakes, each month's change in LST is the area-weighted average of all CP4A lakes within the R25 lake.

**Observations**. We use two satellite-derived products: the Tropical Rainfall Measuring Mission 3B42 product, version 7 (TRMM[20]) and the bias-corrected version

of the the CPC MORPHing technique data (CMORPH[21]). TRMM and CMORPH are available at 3-hourly time resolution and a maximum horizontal resolution of 0.25°. Both products are derived from a combination of infrared and microwave sounders and calibrated against gauge data. CMORPH uses the same set of passive microwave instruments as TRMM. However, it is different in that it propagates these precipitation estimates using motion vectors, which are obtained from infrared data from geostationary satellites. In CMORPH, bias correction over land is done by matching probability density functions against daily NCEP CPC gauge analysis using optimal interpolation with orographic correction, whilst in TRMM, each 3-hourly field is scaled to sum to the corresponding monthly GPCC gauge field. This bias-corrected CMORPH data give quite different rainfall estimates compared with the original version 1 of the data (CMORPH-v1[49]).

Previous studies evaluating satellite-based precipitation estimates over Africa have shown that TRMM 3B42 and CMORPH both perform well in capturing ground-based observations of rainfall on seasonal and annual timescales[50]. On shorter timescales, TRMM rainfall is somewhat more intermittent than CMORPH. Over Africa, TRMM and CMORPH-v1 have been found to have quite different characteristics in terms of day-to-day variability[3]. These differences are likely related to the different satellite data sources, algorithms and bias correction techniques used in each case. It is known that both datasets tend to underestimate smaller daily rainfall totals and can overestimate larger ones[51]. The bias-corrected CMORPH gives a better representation of precipitation in terms of spatial patterns and daily variability compared with raw CMORPH data over a global domain; and globally, it is found to be better than TRMM in representing 3 hourly and daily variability in warm season precipitation, but shows a worse bias during cold seasons[21]. For mesosites in Africa (Mali, Niger and Benin), CMORPH gives very good agreement with AMMA–CATCH that is better than TRMM. However, near Guinea, there is a region of anomalous low precipitation in the bias-corrected

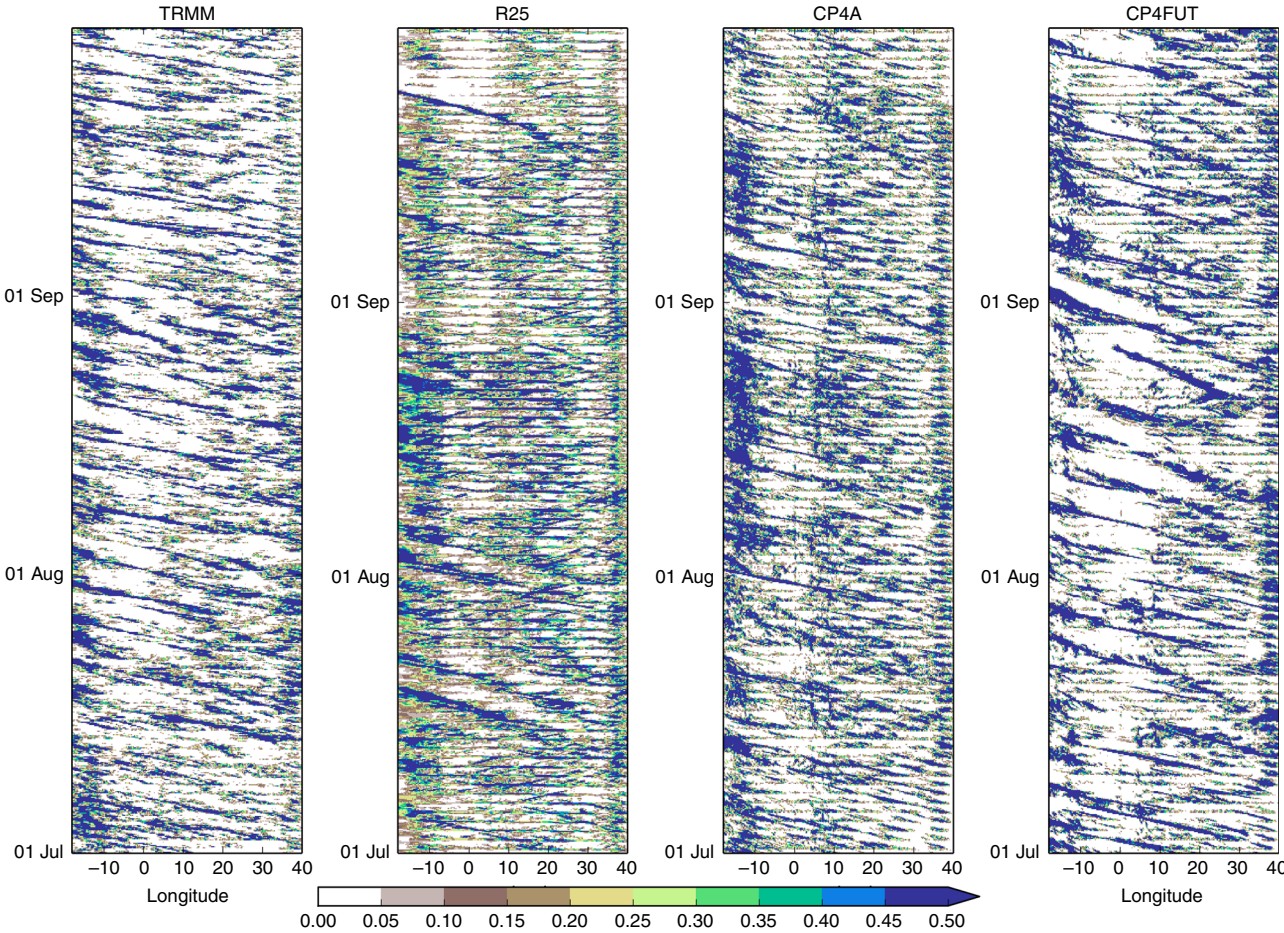

**Fig. 9** Hovmöller plots of 3-hourly precipitation. Precipitation (mm h$^{-1}$) averaged over the latitude band 5–15° N for July–August–September (JAS) 2001 for TRMM, R25 and CP4A and for the equivalent season in the CP4A future simulation

CMORPH (visible in Fig. 2) not seen in other datasets (TRMM, GPCC or GPCP[52]) thus raising questions about the reliability of this feature, which originates from the bias-correction step (this feature is not seen in CMORPH-v1).

Overall, previous studies indicate that both TRMM and CMORPH give a valid representation of rainfall over Africa, although with differences especially on shorter timescales. There is some suggestion that the bias-corrected CMORPH may be more reliable in some regions, but there are some anomalous features and this is a relatively new product with limited evaluation to date. In particular, CMORPH-v1 may be more reliable than CMORPH for the Gulf-of-Guinea region.

In this study, we compare model precipitation with both TRMM and CMORPH (and also CMORPH-v1 for the sub-region analyses, as this may be more reliable for the Gulf-of-Guinea region) to get an indication of observational uncertainty. Consistent with previous studies, we find that it rains less frequently in TRMM than CMORPH, although the observational uncertainty is considerably smaller than the R25-CP4A model differences (Fig. 3). Rainfall intensity is higher in TRMM than CMORPH; and the R25 negative bias in rainfall intensity is reduced compared with CMORPH, but is still present (Fig. 4). Thus the improvement in the representation of rainfall intensity, occurrence and extremes in CP4A compared with R25 is robust to the choice of observational dataset.

**Analysis methods and signficance testing**. All analysis here is carried out at the 25-km scale, with the 3-hourly precipitation fields from both models being aggregated onto the 0.25° observational grid using area-weighted regridding.

The bootstrap resampling[53] is used to estimate uncertainties in model biases or future changes due to natural variability. The resampling is done in yearly blocks, to account for temporal correlation in the precipitation data, so that we only assume independence in hourly rainfall for a given season between years. For the map plots, where resampling was done at each grid point, a total of $n = 100$ bootstrap samples were produced by selecting 10 years from the full dataset (either the observations, the present-day or future model simulation) with replacement. For the regional analyses, a larger bootstrap of $n = 1000$ could be afforded. These bootstraps are used to produce $n$ estimates of the difference between the model and the observations, or between the future and present-day runs, allowing a confidence interval for the difference to be calculated.

Probability values (p-values) can be estimated from the bootstrap, which allow the significance of the difference to be computed. In particular, the p-value is the probability of finding the given difference when the null hypothesis (of zero difference) is true. A small p-value is strong evidence to reject the null hypothesis. For the map plots, p-values were estimated at each grid-point as follows. Firstly, the metric of interest (i.e., the test statistic which is the model bias or future change in a given precipitation metric) was calculated $n$ times from the bootstrap. The mean of the bootstrapped metric was then computed, and subtracted from each $n$ bootstrap estimate; this creates a $n$ number of zero-centred metrics and gives us an estimate of the probability distribution of the test statistic under the null hypothesis. Finally, the original metric is then compared with this null distribution, and the p-value is estimated based on which quantile the original metric corresponds to relative to the null distribution. For instance, if the original metric is below 2% or above 98% of the values in the simulated null distribution, the uncorrected p-value would be 0.02.

Note the above assumes that the probability distribution of the metric is independent of its mean and can be translationally moved. The smallest p-value that the above can yield is $\frac{1}{n}$ when the actual metric lies outside the range of the bootstrapped null distribution; for $n = 100$ this corresponds to 1% and is sufficient for significance testing at the 5% level used here for map plots. For maps showing the ratio of future/present-day frequency of exceeding a present-day extreme threshold, the test statistic is the difference between the present-day and future frequency (not their ratio).

When carrying out field significance tests for map plots, one would expect some significant results to occur by chance[54,55]. The problem is further complicated by the natural spatial correlation of geophysical data, which leads to incorrect identification of significant results[55]. To address this problem, we apply a correction to the p-value[55], thereby controlling the false discovery rates in multi-hypothesis testing[56]. The corrected p-values are then compared with a revised global significance level which is 2× the level for non-field significance tests[55]. As the tests here are two-tailed, the non-field significance level has to be halved, which cancels the 2× multiplier. The p-value corrections are implemented in various open-source numerical analysis software: *p.adjust* in the R package *stats* and *stats.multitest* in the Python package *statsmodels*.

The above bootstrap resampling estimates uncertainty due year-to-year variability. We note that the influence of multi-decadal variability on the results is

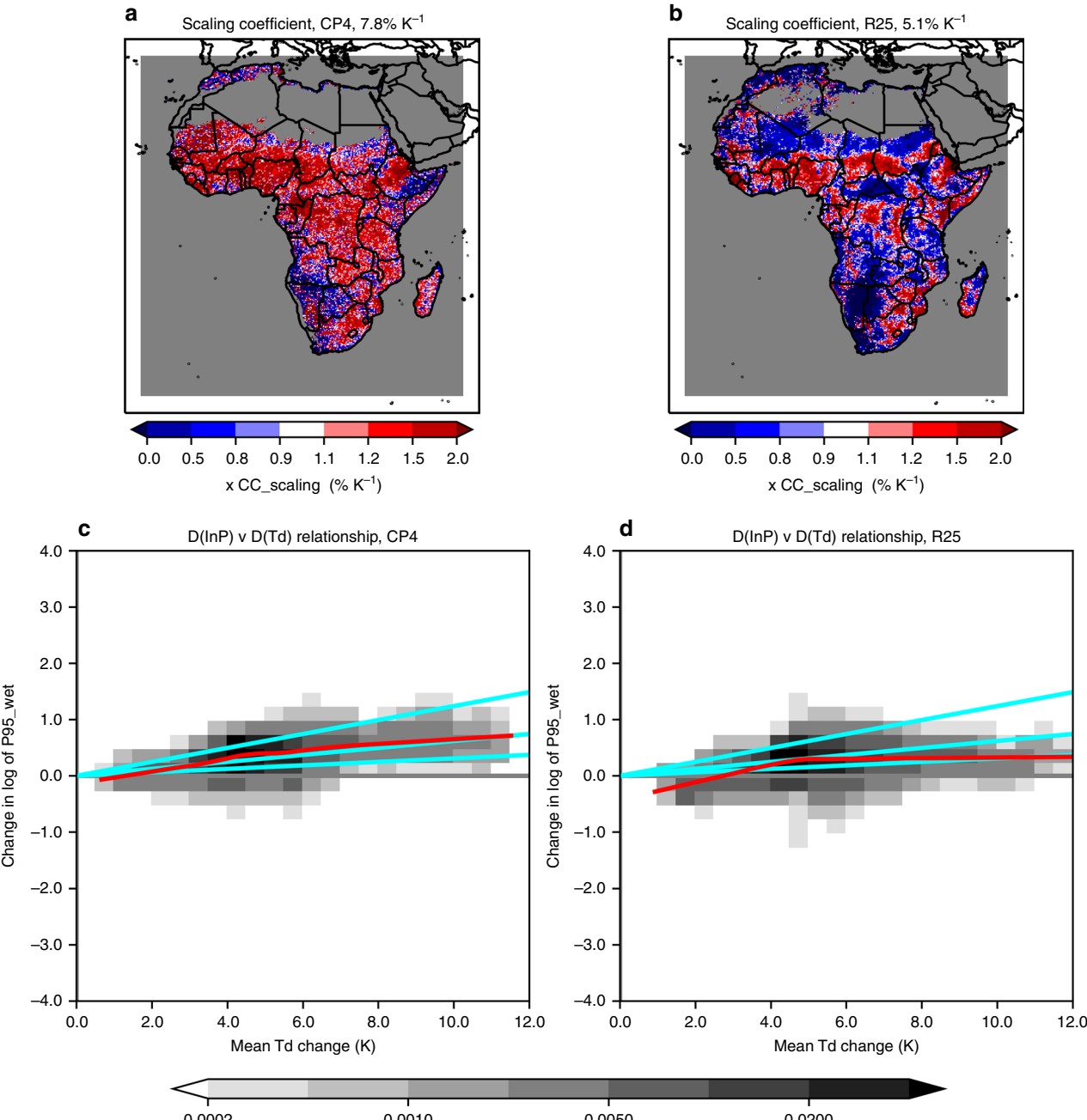

**Fig. 10** Scaling between future changes in extreme 3-hourly precipitation intensity and the dew point temperature. **a, b** Scaling coefficient given by future change in logarithm of extreme precipitation intensity divided by future change in mean dew point temperature, for the wet season, for CP4A and R25. The median scaling across Africa is indicated in the panel titles; colours correspond with the scaling coefficient divided by the Clausius–Clapeyron relationship of 6.2%K$^{-1}$ (such that a value of 1 corresponds to CC scaling). **c, d** Joint probability distribution of change in logarithm of extreme precipitation intensity versus change in mean dew point temperature $T_d$, for the wet season, across Africa, for CP4A and R25. Cyan lines show the relationship for 0.5, 1 and 2 times CC-scaling; red lines show the average relationship obtained from fitting a Lowess regression line. Extreme precipitation intensity is defined as the 95th percentile of wet values (>0.1 mm h$^{-1}$), for daily maximum 3-hourly precipitation, and is set to missing (and masked in grey) over sea points and where less than 5% of the data is wet. The wet season is the 3-month period with the highest mean precipitation in TRMM, defined on a grid-point basis

small due to our adopted treatment of SSTs. In particular, as outlined above, the future SSTs are configured as a time-invariant delta (given by the 30 year mean SST change for each month) applied to the present-day time-varying SSTs, and human induced warming is expected to dwarf any influence of natural climate variability on the 30-year mean change.

**Clausius–Clapeyron scaling.** The thermodynamic Clausius–Clapeyron (CC) relation is given by:

$$\frac{\partial e_s}{\partial T} = \frac{L_v}{R_v T^2} e_s \tag{1}$$

where $T$, $e_s$, $L_v$ and $R_v$ are temperature, saturation vapour pressure of water, vaporisation enthalpy of water vapour and the water vapour gas constant. In general, $L_v$ is temperature-dependent, but the temperature dependency is small for typical lower tropospheric temperatures.

Assuming temperature changes ($\Delta T$) are small relative to its absolute value $\left(\frac{\Delta T}{T} \ll 0.1\right)$, the climate change relationship between temperature and precipitation is:

$$\frac{\Delta P}{P} \approx \frac{\Delta e_s}{e_s} \approx \frac{L_v}{R_v T^2} \Delta T = \gamma \Delta T \tag{2}$$

in which $\gamma$ the (CC scaling factor) is:

$$\gamma = \frac{L_v}{R_v T^2} \qquad (3)$$

$\gamma$ is about 6.2%$K^{-1}$ for typical surface air temperatures (23 °C) over Africa. Under the assumption of constant relative humidity, and providing storm dynamics do not change, changes in extreme precipitation intensity are expected to increase at this rate of 6.2% per degree of warming[33]. However, changes in moisture availability may be much less than this temperature-dependent maximum i.e., relative humidity may decrease. To account for this, we use near-surface dew point temperature as the scaling variable. Dew point temperature ($T_d$) is the temperature to which an air parcel must be cooled to reach saturation, and is a measure of specific humidity translated to temperature using the CC relationship:

$$e = e_0 \exp\left[\frac{L_v}{R_v}\left(\frac{1}{T_0} - \frac{1}{T_d}\right)\right] \qquad (4)$$

where $e$ is water vapour pressure, $e_0$ is 0.611kPa and $T_0$ is 273 K. Given the definition of relative humidity RH (in %) and $e_s$:

$$\text{RH} = 100\frac{e}{e_s} \qquad (5)$$

$$e_s = e_0 \exp\left[\frac{L_v}{R_v}\left(\frac{1}{T_0} - \frac{1}{T}\right)\right] \qquad (6)$$

it is possible to calculate $T_d$:

$$\frac{1}{T_d} = \frac{1}{T} + \frac{R_v}{L_v}\ln\left(\frac{100}{\text{RH}}\right) \qquad (7)$$

## Data availability

The CP4A and R25 datasets generated under the FCFA IMPALA project, and analysed in the current study, will be publicly available from July 2019 from the Centre for Environmental Data Analysis (CEDA) archive (http://archive.ceda.ac.uk/). Until then, these data are available on request from Cath Senior (Chair of the IMPALA CP4A Working Group), but restrictions apply to the use of these data, which are used under license and must respect the work plans of the FCFA regional consortia.

## Code availability

Data processing scripts are available from the corresponding author upon request.

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

## Acknowledgements

This work was funded by the Department for International Development/Natural Environment Research Council via the Future Climate for Africa (FCFA) funded project Improving Model Processes for African Climate (IMPALA, NE/M017214/1 and NE/M017230/1). Funding was also provided via the FCFA HyCRISTAL (NE/M019985/1, for Kendon, Marsham and Rowell) and AMMA-2050 (NE/M019977/1, for Kendon, Marsham, Berthou and Rowell) projects. E. Kendon and S. Berthou gratefully acknowledge funding from the Joint UK BEIS/Defra Met Office Hadley Centre Climate Programme (GA01101). Thanks to Declan Finney for useful discussions on CC scaling, and Steven Chan for code and advice on significance testing.

## Author contributions

E.J.K. helped set up the model experiments, carried out the analysis and wrote the paper. R.A.S. and S.T. carried out the CP4A and R25 present-day and future experiments, respectively, and R.A.S. analysed vertical stability profiles and advised on the differing convective responses in the two models. J.H.M. advised on the scaling analysis and contributed extensively to the paper. S.B. helped with python analysis code and D.P.R. helped design the set up of the future CP4A experiment. All authors commented on the paper.

## Additional information

**Competing interests:** The authors declare no competing interests.

