## [Peer Review File · Nature Communications]

The manuscript "Enhanced future changes in wet and dry extremes over Africa at convection-permitting scale" by Kendon et al. investigates how climate change might affect precipitation over Africa by using a 25 km and 4.5 km model. The latter is explicitly simulating deep convection while the former needs a deep convection parameterization. The authors conclude that the 4.5 km model is able to more realistically simulate heavy precipitation events and that this model shows a stronger increase in heavy rainfall and longer dry spells at the end of the century under the RCP8.5 pathway. The manuscript improved since the first submission to Nature Climate Change and I appreciate that the authors addressed many of my initial comments. In particular, the authors added more process-based analysis that aims to explain why the 4.5 km model shows larger changes in the future. I still think that a more careful wording is needed concerning the "beyond expectation" increase in heavy precipitation rates in the 4.5 km model unless the authors can show that the 7.8 % increase per degree warming is significantly different from the expected 6-7 % increase according to Clausius-Clapeyron. This is particularly important since the authors are not able to sufficiently sample natural climate variability in two 10-year long simulations. I, therefore, suggest to rephrase your manuscript and focus more on the fact that the 4.5 km model shows more extreme climate change signals compared to the 25 km model rather than focusing on the amplitude of your changes (e.g., if 7.8% is beyond expected changes). A more detailed explanation about these points and other comments can be found below.

General comments:

- Generally, I would suggest being careful by inferring that an increase in extreme precipitation due to climate change directly relates to increases in flood frequency and flood intensity. Sharma et al. (2018) demonstrate that the two are not directly linked and that additional changes such as changes in antecedent soil moisture have to be considered.

- You state that extreme precipitation is increasing and that dry spells will be longer in the future. How does that relate to average precipitation?

- A discussion of historic changes in observed precipitation in Africa would be beneficial (e.g., Taylor et al. 2013; Maidment et al. 2015). Natural climate variability plays a major role in parts of Africa. E.g., Taylor et al. 2017 showed that the frequency of extreme Sahelian storms tripled since 1982 in satellite observations. Detecting the climate change impact on extreme precipitation in a region that has such a large internal variability with two 10-year long simulations is very challenging if not impossible. A thorough discussion about the impact of internal variability on your results is therefore essential.

Specific comments:

P2L11-13 & P6L19 & P7L4-8: Are the 7.8 % increase in extreme precipitation statistically significantly higher than the expected 6-7 % per degree warming? The authors make this claim in the abstract, in the conclusions, and at several other areas in the paper. Li et al. (2018) show that even a record length of 150 years is too short to estimate precipitation scaling rates reliably and that a multi-model ensemble is required to calculate meaningful statistics. I do not doubt that the authors see a statistically significant increase in future extreme precipitation but I question if

you can significantly differentiate a 0.8 % increase from the large background noise with simulating two 10-year long time periods. Also, previous studies showed that the scaling rates are strongly dependent on how you define extremes with larger scaling rates for higher percentiles (e.g., Ban et al. 2015, Pendergrass 2018). Adding a plot similar to Fig. 4 Ban et al. (2015; see below) would be very interesting.

Figure 4. Projected changes in the precipitation at high percentiles on hourly time scales in summer. Shown are relative change in high percentiles of precipitation normalized by the local temperature change (in percent per degree warming). The results are based on the spatial average of the grid point relative changes and scaling rates (adapted from Ban et al. 2015).

P6L25: Similar to above; is the 11 % result statistically significant? Convection-permitting simulations are better in representing the spatial structure of convective storms whereas a 25 km model typically spatially smooths the area of heavy precipitation. Extreme precipitation is typically very localized and displacements of extreme convective storms in the future compared to the current climate can easily cause very large scaling rates on a grid cell level especially if you only simulate 10 years in each climate. These displacements do not necessarily mean that you simulate super CC scaling but that you are undersampling extreme convective storms. Please provide the number of grid cells that have very low/negative scaling rates in addition to grid cells that have 2xCC scaling. Negative scaling rates typically happen due to the above-mentioned displacements. In general, I would recommend avoiding grid scale statistics when it comes to scaling rates unless you have a very long record to select from or you average over a larger pool of data (e.g., 20x20 grid cells).

P3L34: What do you mean by ‘capture the balance’?

P4L1: Does this mean shifted by one month?

P5L8: Do the up to 8x mean that 60 mm/3h events are ~70% more frequent in the 25 km model? Also, how does this statistic change if you use the same percentile value (percentile of the 60 mm/3h in 4.5 km model) in the 25 km model?

P5L8-12: The Dakar flooding in 2012 is a good example of an extreme precipitation event in Africa but your threshold of 60 mm/3h is way lower than the flood-producing event that had 144 mm/h. I would be careful in calling 60 mm/3h events flood producing. Can you say anything about changes in events on the order of 150 mm/3h from your simulations?

P5L31: There are some eastward propagating features in TRMM as well.

P5L36: This result agrees well with the findings by Rasmussen et al. 2017.

Literature:

Ban, N., Schmidli, J. and Schär, C., 2015. Heavy precipitation in a changing climate: Does short-term summer precipitation increase faster?. *Geophysical Research Letters*, 42(4), pp.1165-1172.

Li, C., Zwiers, F., Zhang, X. and Li, G., How much information is required to well-constrain local estimates of future precipitation extremes?. *Earth's Future*.

Sharma, A., Wasko, C. and Lettenmaier, D.P., 2018. If Precipitation Extremes Are Increasing, Why Aren't Floods?. *Water Resources Research*, 54(11), pp.8545-8551.

Pendergrass, A.G., 2018. What precipitation is extreme?. *Science*, 360(6393), pp.1072-1073.

Rasmussen, K.L., Prein, A.F., Rasmussen, R.M., Ikeda, K. and Liu, C., 2017. Changes in the convective population and thermodynamic environments in convection-permitting regional climate simulations over the United States. *Climate Dynamics*, pp.1-26.

Maidment, R.I., Allan, R.P. and Black, E., 2015. Recent observed and simulated changes in precipitation over Africa. *Geophysical Research Letters*, 42(19), pp.8155-8164.

Taylor, R.G., Todd, M.C., Kongola, L., Maurice, L., Nahozya, E., Sanga, H. and MacDonald, A.M., 2013. Evidence of the dependence of groundwater resources on extreme rainfall in East Africa. *Nature Climate Change*, 3(4), p.374.

Taylor, C.M., Belušić, D., Guichard, F., Parker, D.J., Vischel, T., Bock, O., Harris, P.P., Janicot, S., Klein, C. and Panthou, G., 2017. Frequency of extreme Sahelian storms tripled since 1982 in satellite observations. *Nature*, 544(7651), p.475.

Reviewer #2 (Remarks to the Author):

This is my second reading of this manuscript, as I had already reviewed the version of this paper submitted to Nature Climate Change.

At that time, I had appreciated the work, recognizing that convective scale simulations are really needed to better understand present climate characteristics and its future evolution especially for regions such as tropical Africa.

My main criticism, i.e. the too short duration of the simulation, has somehow been addressed, as, now, 10 year simulations have been used. Although this duration is still relatively short to fully grasp all the natural variability and associated uncertainties (both in the present and in the future), I do acknowledge that long term (centennial transient) convection-permitting continental-scale simulation will not be available for a while. So, to this regard, this work is indeed innovative and important.

I do also appreciate the effort of the authors to address my other points regarding clarifying part of the text and methodology and to revise the supporting literature, that is more complete now.

The emphasis of dry spell duration in this new version is also interesting; although the fact that increasing model resolution (and better representation of physical process) will increase the occurrence and intensity of extreme events is not really completely new, I found interesting for instance, from Supplementary Figs 3 and 4 that CP4 projects, compared to R25 a much shorter but more intense wet season over most of tropical Africa, (which again confirms previous GCMs and RCMs based studies) that could be possibly emphasized more for its obvious potential impact on society and economy of Africa.

Reviewer response document

Manuscript: NCOMMS-18-34771-T

Title: Enhanced future changes in wet and dry extremes over Africa at convection-permitting scale

Please find below a point-by-point response to the reviewers' comments, with responses shown in red text. A marked up copy of the manuscript (kendon18impala_revised_markedup.pdf) showing changes to the paper is included in the submission. In this changes made in response to the reviewers' comments are shown in red; editorial changes to meet *Nature Comms* formatting requirements (in particular text moved from the Supplementary Material to the main text and minor changes to the text to accompany the movement of Supplementary Figures) are shown in blue.

Reviewer responses

Reviewer #1

The manuscript "Enhanced future changes in wet and dry extremes over Africa at convection permitting scale" by Kendon et al. investigates how climate change might affect precipitation over Africa by using a 25 km and 4.5 km model. The latter is explicitly simulating deep convection while the former needs a deep convection parameterization. The authors conclude that the 4.5 km model is able to more realistically simulate heavy precipitation events and that this model shows a stronger increase in heavy rainfall and longer dry spells at the end of the century under the RCP8.5 pathway. The manuscript improved since the first submission to Nature Climate Change and I appreciate that the authors addressed many of my initial comments. In particular, the authors added more process-based analysis that aims to explain why the 4.5 km model shows larger changes in the future.

Thank you for your original comments on the manuscript, they helped to improve it substantially. We believe that the additional process-based analysis provides insight into why the 4.5km model shows larger changes, which is important for our confidence in the results.

I still think that a more careful wording is needed concerning the "beyond expectation" increase in heavy precipitation rates in the 4.5 km model unless the authors can show that the 7.8 % increase per degree warming is significantly different from the expected 6-7 % increase according to Clausius-Clapeyron. This is particularly important since the authors are not able to sufficiently sample natural climate variability in two 10-year long simulations. I, therefore, suggest to rephrase your manuscript and focus more on the fact that the 4.5 km model shows more extreme climate change signals compared to the 25 km model rather than focusing on the amplitude of your changes (e.g., if 7.8% is beyond expected changes).

Future changes in extreme precipitation intensity are significantly higher in CP4A than in R25 compared to year-to-year natural variability over much of central Africa, in the region where CP4A shows super-CC scaling (see Figure A2 below). However, we acknowledge the

reviewer's concern that 10 years of model data is not sufficient to fully characterise extremes, and hence scaling, at the grid point scale. This is now stated in the text. We have also rephrased the manuscript removing statements that increases in extreme-precipitation "are beyond expectations from simply scaling increased atmospheric moisture with warming". Instead we now focus on the fact that scaling rates are higher in CP4A compared to R25, rather than the extent to which they actually exceed the 6-7%/K CC-relationship.

Please see response to specific comment on this below, for detail on changes made to the text.

A more detailed explanation about these points and other comments can be found below.

General comments:

- Generally, I would suggest being careful by inferring that an increase in extreme precipitation due to climate change directly relates to increases in flood frequency and flood intensity. Sharma et al. (2018) demonstrate that the two are not directly linked and that additional changes such as changes in antecedent soil moisture have to be considered.

We fully acknowledge that there is not a one-to-one correspondence between extreme precipitation and flooding. There are many other contributing factors to flooding, such as antecedent conditions, land use and flood management measures. For example Tarhule (2005), which is now cited in the manuscript, gives examples of flooding over the Sahel primarily driven by land use patterns and also discusses the importance of cumulative rainfall in the days prior to the heavy event. Having said that, however, short-duration intense rainfall is the primary driver of flash flooding. In this paper, it is this type of rainfall that we are primarily focussing on. In particular, we are looking at changes in 3hrly precipitation extremes, rather than multi-day precipitation extremes that are likely to be more important for fluvial flooding.

In the paper, we only mention flooding in the paragraph where we discuss the Dakar flooding event in 2012, and here we are careful about inferring changes in flooding from extreme precipitation changes. In particular we have changed the text to say that: "These relatively rare events *may* be flood-inducing" and "such a high accumulation over the 25km scale may lead to *local flash flooding in some cases*". We have added reference to Tarhule et al (2005).

Despite the difficulties in inferring changes in flooding from extreme precipitation changes, we believe that the brief discussion of the Dakar flood event and its impacts is useful in helping readers appreciate the important implications of greater increases in short-duration precipitation extremes found at convection-permitting scale.

Tarhule A. (2005) Damaging rainfall and flooding: the other Sahel hazards. *Climatic Change* 72: 355–377 DOI: 10.1007/s10584-005-6792-4

- You state that extreme precipitation is increasing and that dry spells will be longer in the future. How does that relate to average precipitation?

We have moved Supplementary Fig 2 showing mean precipitation changes to the main paper (now Fig. 2). From Fig 2 it can be seen that mean wet season precipitation shows modest increases in both models. The underlying changes in precipitation intensity and occurrence are larger, particularly in the convection-permitting model, and to some extent cancel out in the mean precipitation change.

This is already stated in the text:

“Compared to R25, CP4A shows slightly smaller increases in mean precipitation; but importantly CP4A shows much larger decreases in rainfall occurrence (18% compared to 1.5% in R25) and larger increases in precipitation intensity (39% compared to 22%) ...”

We have added further text discussing mean precipitation changes, specifically over the Congo where CP4A shows a smaller increase in mean precipitation compared to R25: “CP4A’s smaller increase in mean precipitation over the Congo, corresponds to its greater reduction in rainfall frequency there. These differences between CP4A and R25 show how changing convection affects much larger scales.”

Given the importance of the underlying changes, and the fact that these are significantly different at convection-permitting scale, we have also moved the figures showing these results from the Supplementary Material (previously Supplementary Figs. 3 and 4) to the main text (now Figs. 3 and 4).

- A discussion of historic changes in observed precipitation in Africa would be beneficial (e.g., Taylor et al. 2013; Maidment et al. 2015). Natural climate variability plays a major role in parts of Africa. E.g., Taylor et al. 2017 showed that the frequency of extreme Sahelian storms tripled since 1982 in satellite observations. Detecting the climate change impact on extreme precipitation in a region that has such a large internal variability with two 10-year long simulations is very challenging if not impossible. A thorough discussion about the impact of internal variability on your results is therefore essential.

We fully acknowledge that natural climate variability plays a major role in historic precipitation changes, however, we dispute that detecting the climate change impact on precipitation is “impossible”. In particular here we are considering changes in precipitation at the end of the century, for which there is a higher signal to noise ratio than during the observed period. In this case, changes in extreme precipitation are clearly significant compared to natural variability sampled by the 10-year model simulations, with the changes largely explicable in terms of increasing atmospheric moisture.

We appreciate that with 10-year long simulations it is not possible to sample uncertainty due to multi-decadal natural variability. However as noted in the Methods section “the influence of multi-decadal variability on the results is small due to our adopted treatment of SSTs. In particular, ... the future SSTs are configured as a time-invariant delta (given by the 30 year mean SST change for each month) applied to the present-day time-varying SSTs, and human induced warming is expected to dwarf any influence of natural climate variability on the 30 year mean change.”

We have added a discussion of the impact of internal variability to the Discussion section: “Recent studies assessing historic changes in observed precipitation have shown that natural variability plays a major role in parts of Africa (Maidment et al 2015); although recent increases in rainfall over the Sahel have been linked to increased greenhouse gases (Dong and Sutton, 2015). Results here are based on single 10-year model realisations of climate change, nevertheless, we are able to detect clear changes in extreme precipitation at the end of the century above year-to-year natural variability. This is due to the fact that the signal of change is much larger by 2100, with changes in extreme precipitation largely explicable in terms of increasing atmospheric moisture with warming. There is no influence of oceanic decadal variability on the results due to our adopted treatment of sea-surface temperatures (see Methods). The relatively short simulations here, however, do limit our sampling of extreme events and hence our ability to robustly characterise changes in the extreme tail of the precipitation distribution at the local scale.”

Taylor et al (2017) is referenced in the new Discussion section, where we discuss the impact of changes in wind shear for the intensification of storms:
“... increased wind shear favours the mesoscale organisation of convection and this has led to a recent intensification of extreme Sahel storms (Taylor et al, 2017).”

Maidment et al (2015) and Dong and Sutton (2015) are now referenced. Maidment et al (2015) finds robust regional increases in rainfall over the Sahel (29 to 43 mm yr⁻¹ per decade) and Southern Africa (12 to 41 mm yr⁻¹ per decade) and drying over East Africa (-14 to -65 mm yr⁻¹ per decade in March–May rainfall) in recent decades (1983-2014). Increases in southern Africa rainfall are thought to be due to strengthening of the Walker circulation and relate to internal climate variability; whilst increases in Sahel rainfall have been linked to increased greenhouse gases (Dong and Sutton, 2015). Over Central Africa there is considerable uncertainty in trends (in both magnitude and sign).

Specific comments:

P2L11-13 & P6L19 & P7L4-8: Are the 7.8 % increase in extreme precipitation statistically significantly higher than the expected 6-7 % per degree warming? The authors make this claim in the abstract, in the conclusions, and at several other areas in the paper. Li et al. (2018) show that even a record length of 150 years is too short to estimate precipitation scaling rates reliably and that a multi-model ensemble is required to calculate meaningful statistics. I do not doubt that the authors see a statistically significant increase in future extreme precipitation but I question if you can significantly differentiate a 0.8 % increase from the large background noise with simulating two 10-year long time periods. Also, previous studies showed that the scaling rates are strongly dependent on how you define extremes with larger scaling rates for higher percentiles (e.g., Ban et al. 2015, Pendergrass 2018). Adding a plot similar to Fig. 4 Ban et al. (2015; see below) would be very interesting.

Figure 4. Projected changes in the precipitation at high percentiles on hourly time scales in summer. Shown are relative change in high percentiles of precipitation normalized by the local temperature change (in percent per degree warming). The results are based on the spatial average of the grid point relative changes and scaling rates (adapted from Ban et al. 2015).

Figure A1: Scaling rate across Africa (%/K) as a function of wet-value percentile. Plotted is the median scaling rate across Africa, with the shaded region showing the interquartile range in spatially-varying scaling rates. Results are shown for percentiles of daily maximum 3-hourly precipitation, for wet values ($>0.1 \text{ mm h}^{-1}$) only. Scaling at each grid point is given by the future change in the logarithm of precipitation, for the given percentile, divided by the future change in mean dew point temperature, for the wet season locally. Local values are masked where less than 5% of the data is wet (see Fig. 10 in main paper). The Clausius-Clapeyron relationship of 6.2%/K is shown in black.

The above Figure (A1) showing the variation in the scaling rate across Africa as a function of the wet-value percentile has been added to the Supplementary Material. This shows larger scaling rates for higher percentiles, consistent with Ban et al 2015. The median scaling rate across Africa is higher than CC-scaling in CP4A for all percentiles shown; whilst median scaling rates are below CC-scaling in R25 for wet-value percentiles below 99th, but above CC-scaling for higher percentiles. In all cases, the median scaling rate is higher in CP4A than R25. There is considerable spatial variability in the scaling rate (indicated by the shaded region in the above Figure), with both models showing super-CC scaling locally, although with a higher proportion of grid points showing super-CC scaling in CP4A.

We have assessed whether future changes in extreme precipitation intensity (in this case defined as the 99th percentile of wet values for 3-hourly precipitation, as in Fig. 5) are significantly different between the CP4A and R25 models at the grid point scale. This shows that future changes in extreme precipitation intensity are significantly higher at the 5% level in CP4A compared to those in R25 across much of central Africa (Figure A2 below). Thus in the region across Africa where CP4A shows super-CC scaling in Fig. 10, changes in

extreme precipitation intensity are significantly higher in CP4A. Over Northern Africa, we do not calculate scaling due to too few wet values; and over the Horn of Africa and southern Africa, where changes are not significantly different between CP4A and R25 in Fig A2, CP4A largely shows negative scaling (Fig. 10). Overall these results suggest that the higher scaling rates seen in CP4A compared to R25 over much of central Africa, are associated with significantly greater increases in extreme precipitation intensity in CP4A.

CP4 change minus R25 change, pw99

Figure A2: Difference in future changes in extreme precipitation intensity between CP4A and R25. Differences are masked in white where they are not significant at the 5% level compared to year-to-year variability. Extreme precipitation intensity is defined as the 99th percentile of wet values ($>0.1 \text{ mm h}^{-1}$), for 3-hourly precipitation during the wet season. The wet season is the 3 month period with the highest mean precipitation in TRMM, defined on a grid-point basis.

In the manuscript text we have made changes to include discussion of the above results (see below). In particular we now emphasise the spatial variability in the scaling rates, and discuss that this may be symptomatic of an under-sampling of extreme precipitation in the 10-year model simulations. We note that future changes in extreme precipitation intensity are significantly higher in CP4A than in R25 compared to year-to-year natural variability over much of central Africa, in the region where CP4A shows super-CC scaling. However, we acknowledge the reviewer's concern that 10 years of model data is not sufficient to fully characterise extremes, and hence scaling, at the grid point scale.

"In both models, the scaling rate increases for higher percentiles of the precipitation distribution (Supplementary Fig. 13), consistent with previous studies (Ban et al 2015), with the scaling rate consistently higher in CP4A than R25 ...

"We see large departures from CC-scaling locally, with evidence of 'super-CC scaling' occurring more widely across Africa in CP4A (Fig 10, Supplementary Fig. 13). In particular,

11% of 25km grid points show scaling coefficients $>2xCC$ in CP4A compared to 4% in R25. Both models also sample negative scaling rates, with 5% of grid points showing negative scaling in CP4A and 13% in R25. Locally varying negative to strongly-positive scaling rates typically happen due to displacements of extreme convective storms in future, and may be symptomatic of an undersampling of extreme storms in the 10-year model simulations. The fact that super-CC scaling is more prevalent than negative scaling in CP4A (but not R25) shows that this is not simply explained by displacements of extreme storms. We note that over much of central Africa where super-CC scaling is observed in CP4A, future changes in extreme precipitation intensity (Fig. 5) are significantly higher in CP4A than in R25 taking into account year-to-year variability. These results suggest that higher scaling rates at convection-permitting scale for Africa as a whole are robust, but given the results are based on single 10-year model realisations, there is uncertainty in the actual scaling values at the 25km grid point scale.”

We have also changed the abstract, removing the statement that increases in extreme-precipitation “are beyond expectations from simply scaling increased atmospheric moisture with warming”. We have also similarly removed this statement from the conclusions, and instead in the Discussion section now focus on the fact that scaling rates are higher in CP4A compared to R25, rather than the extent to which they actually exceed the 6-7%/K CC-relationship.

We have added reference to Ban et al 2015:

Ban, N., Schmidli, J. and Schär, C., 2015. Heavy precipitation in a changing climate: Does shortterm summer precipitation increase faster?. *Geophysical Research Letters*, 42(4), pp.1165-1172

P6L25: Similar to above; is the 11 % result statistically significant? Convection-permitting simulations are better in representing the spatial structure of convective storms whereas a 25 km model typically spatially smoothest the area of heavy precipitation. Extreme precipitation is typically very localized and displacements of extreme convective storms in the future compared to the current climate can easily cause very large scaling rates on a grid cell level especially if you only simulate 10 years in each climate. These displacements do not necessarily mean that you simulate super CC scaling but that you are undersampling extreme convective storms. Please provide the number of grid cells that have very low/negative scaling rates in addition to grid cells that have $2xCC$ scaling. Negative scaling rates typically happen due to the above-mentioned displacements. In general, I would recommend avoiding grid scale statistics when it comes to scaling rates unless you have a very long record to select from or you average over a larger pool of data (e.g., 20x20 grid cells).

We first note that we analyse CP4A on the 25km scale and not the 4.5km grid point scale. The percentage of 25km grid points showing $>2CC$ scaling is 11% in CP4A compared to 4% in R25; whilst the percentage of grid points showing negative scaling is 5% in CP4A compared to 13% in R25. (These are the results for changes in extreme precipitation intensity, defined as the 95th percentile of wet values of daily maximum 3hrly precipitation). This shows that super-CC scaling in CP4A is not simply explained by displacements of extreme convective storms.

We have added text to the paper discussing these results – quoting statistics for both super-CC scaling and negative scaling, and caution that there may be displacements of extreme convective storms in future, symptomatic of an under-sampling of extreme storms in the 10-year model simulations. We also refer to the new Supplementary figure above which shows the interquartile range in spatially-varying scaling rates:

“We see large departures from CC-scaling locally, with evidence of ‘super-CC scaling’ occurring more widely across Africa in CP4A (Fig 10, Supplementary Fig 13). In particular, 11% of 25km grid points show scaling coefficients $>2xCC$ in CP4A compared to 4% in R25. Both models also sample negative scaling rates, with 5% of grid points showing negative scaling in CP4A and 13% in R25. Locally varying negative to strongly-positive scaling rates typically happen due to displacements of extreme convective storms in future, and may be symptomatic of an undersampling of extreme storms in the 10-year model simulations. The fact that super-CC scaling is more prevalent than negative scaling in CP4A (but not R25) shows that this is not simply explained by displacements of extreme storms.”

We have reproduced the scaling results using pooling of data over 3x3 grid cells (see Figure A3 below). This shows higher scaling rates across Africa in CP4A compared to R25, similar to the result for non-pooled data (c.f. Fig 10 in paper). In this case the statistics for the numbers of 3x3 pooled grid boxes showing super (negative) CC-scaling is 5.8% (2.9%) in CP4A and 2.8% (10.9%) in R25. Thus with the pooled data (as well as the non-pooled data) we find evidence of ‘super-CC scaling’ occurring more widely across Africa in CP4A, whilst negative scaling is much more common in R25. Since the key conclusions are unaffected by pooling, we have decided to show the non-pooled results in the paper for consistency with earlier figures in the manuscript which show changes at the 25km grid-box scale. We do note in the manuscript however that scaling values at the grid point scale are not robust: “...there is uncertainty in the actual scaling values at the 25km grid point scale.”

Figure A3: Scaling between future changes in extreme 3-hourly precipitation intensity and dew point temperature. As Fig 10 in main paper, but for 3x3 pooling of precipitation and temperature data.

P3L34: What do you mean by 'capture the balance'?

Text changed to "captures the variation".

In the future the shift from MAM to OND as the wet season over the Horn of Africa has been shown to relate to the slower retreat of the ITCZ southwards (Dunning et al 2018). This is now stated in the manuscript with a reference to Dunning et al (2018) added.

Dunning, C. M., Black, E. & Allan, R. P. Later wet seasons with more intense rainfall over Africa under future climate change. *J. Climate* (2018).

P4L1: Does this mean shifted by one month?

Yes. We now explicitly say “shifted 1-month later”.

P5L8: Do the up to 8x mean that 60 mm/3h events are ~70% more frequent in the 25 km model? Also, how does this statistic change if you use the same percentile value (percentile of the 60 mm/3h in 4.5 km model) in the 25 km model?

The probability of exceeding 60 mm/3h in the present-day climate is 8x times more frequent in CP4A than R25 for Gulf of Guinea and for East Africa, in their respective wet seasons. This means that such events in R25 are ~13% (=1/8) of the frequency of those in CP4A. For the Sahel, Central Africa and Southern Africa, the present day exceedance rates are more similar, with the CP4A frequency being 30% higher, 3x higher and 10% higher than R25 respectively.

In the future, in CP4A, exceeding 60mm/3h is 7x times more likely compared to the present-day for the Sahel, 4x more likely for the Gulf of Guinea, 6x more likely for Central Africa, 8x more likely for East Africa and 4x more likely for S Africa. By comparison the future changes for R25 are 3x (Sahel), 1x (Gulf of Guinea), 4x (Central Africa), 9x (East Africa) and 5x (S Africa) more likely. Thus importantly although such exceedances are rarer in the present-day in R25 (and thus constitute a more extreme value in the 3hrly precipitation distribution), the future changes are less (Sahel, Gulf of Guinea, Central Africa) or similar (East Africa and S Africa) compared to CP4A.

Here we are discussing events exceeding a specific absolute threshold of 60mm/3h, because of its particular relevance to users. However, we do also consider exceedances of a percentile-based threshold elsewhere in the paper (Figure 6). Given the user-relevance, we feel discussion of future changes in the exceedance of the 60mm/3h threshold is useful. We have added further discussion of these results in the text:

“In CP4A, exceeding 60mm accumulation in 3 hours, at the 25km scale, is 7-8 times more frequent in future compared to the present-day for the Sahel and E-Africa (and 4-6x more frequent in other regions). In these regions, such an event shifts from occurring typically once every 30 years at each grid point in the present-climate to once every 3-4 years in the future. In R25 such exceedances are rarer and typically increase less in future; over East Africa the future increase (of 9x) is similar but such events are 8x rarer than in CP4A.”

P5L8-12: The Dakar flooding in 2012 is a good example of an extreme precipitation event in Africa but your threshold of 60 mm/3h is way lower than the flood-producing event that had 144 mm/h. I would be careful in calling 60 mm/3h events flood producing. Can you say anything about changes in events on the order of 150 mm/3h from your simulations?

The Dakar record of 144 mm/h is for a point measurement, whereas the model results are for a “60 mm accumulation in 3 hours, at the 25km scale”. A value as high as ~150mm/h averaged over a region 25kmx25km does not occur in the model or TRMM/CMORPH observational data. Even as a point measurement, such a value is rare in the observational record and occurred due to the combination of a number of effects occurring coincidentally.

Panthou et al 2014 find an areal reduction factor (ARF) of 0.82 between 25km and point data in the Sahel. Using AMMA-CATCH data, we find that there are 2 events at the 25km scale exceeding 60mm/3h, whilst at the 4.5km scale we need to increase the threshold to 88mm/3h for a similar number of exceedances. This suggests 60mm/3h at 25km is equivalent to 90mm/3h at 4.5km, giving an ARF of 0.67 or even lower between 25km and the point scale. This difference compared to Panthou et al (2014) may be due to more extreme events being more localised.

Using the above ARFs, a value of 144 mm/h at the point scale is estimated to be in the range 90-120mm/h at the 25km scale. The maximum 3hrly precipitation measured in CP4A at the 25km scale anywhere in Africa during the 10 year simulation period (1997-2007) is 80mm/h, which is less than this. Thus we do not appear to sample events as extreme as the Dakar flooding event in the 10-year model simulation.

The threshold we examine of 60mm accumulation in 3 hours corresponds to an event which occurs typically once every 30 years at each grid point in the present-climate in the East Africa and Sahel regions (and more frequently, about once every 7-9 years, in the other regions). Thus this event is relatively rare, but by pooling data over each of the sub-regions, gives a sufficiently large sample size to allow robust statistics. Hence we choose this threshold, rather than a rarer more extreme event for which future changes are likely to be much less robust. Although 60mm/3h is less than the Dakar flooding event, such a high accumulation over the 25km scale is expected to lead to local flash flooding in some cases. For example, Tarhule (2005) reports significant flood damage in Niamey for daily totals of 80mm/day up to 150 mm/day measured at a station point.

We have modified the text where we refer to this event to make the above points, and also have added references to Panthou et al 2014 and Tarhule 2005:

“In CP4A, exceeding 60mm accumulation in 3 hours, at the 25km scale, is 7-8 times more frequent in future compared to the present-day for the Sahel and E-Africa (and 4-6x more frequent in other regions). In these regions, such an event shifts from occurring typically once every 30 years at each grid point in the present-climate to once every 3-4 years in the future. In R25 such exceedances are rarer and typically increase less in future; over East Africa the future increase (of 9x) is similar but such events are 8x rarer than in CP4A. These relatively rare events may be flood-inducing. The highest ever recorded event in Dakar in August 2012, with 144mm in 1h (161mm in 6h) at the local station scale, led to widespread flooding with 287,000 people displaced and 18 deaths (Engel et al 2017). At the 25km scale, the rainfall total is expected to be much less than at the point scale (an areal reduction factor of 0.82 is estimated between point and 25km rainfall data in the Sahel (Panthou et al 2014) but the factor is likely to be lower for localised extreme events). 60mm in 3h is lower than the Dakar flood-producing event, but such a high accumulation over the 25km scale may lead to local flash flooding in some cases (Tarhule, 2005), and is chosen as a user-relevant threshold for which we can estimate more robust statistics of future change.”

G. Panthou, T. Vischel, T. Lebel, G. Quantin, and G. Molinié (2014) Characterizing the space–time structure of rainfall in the Sahel with a view to estimating IDAF curves. *Hydrol. Earth Syst. Sci. Discuss.*, 11, 8409–8441, www.hydrol-earth-syst-sci-discuss.net/11/8409/2014/, doi:10.5194/hessd-11-8409-2014

Tarhule A. (2005) Damaging rainfall and flooding: the other Sahel hazards. *Climatic Change* 72: 355–377 DOI: 10.1007/s10584-005-6792-4

P5L31: There are some eastward propagating features in TRMM as well.

This is now acknowledged:

“There are some eastward moving features in TRMM, but these are less prominent than in R25.”

P5L36: This result agrees well with the findings by Rasmussen et al. 2017.

Rasmussen et al 2017 looking at CPM projections over the US finds that both convective available potential energy (CAPE) and convective inhibition (CIN) increase downstream of the Rockies in a future climate. The increase in CIN suppresses weak to moderate convection and provides an environment where CAPE can build to extreme levels that may result in more frequent severe convection.

These results agree well with the findings here and so reference to Rasmussen et al 2017 has been added to the text:

“Similar results are seen in CPM simulations over the United States (Rasmussen et al 2017).”

Literature:

- Ban, N., Schmidli, J. and Schär, C., 2015. Heavy precipitation in a changing climate: Does shortterm summer precipitation increase faster?. *Geophysical Research Letters*, 42(4), pp.1165-1172.
- Li, C., Zwiers, F., Zhang, X. and Li, G., How much information is required to well-constrain local estimates of future precipitation extremes?. *Earth's Future*.
- Sharma, A., Wasko, C. and Lettenmaier, D.P., 2018. If Precipitation Extremes Are Increasing, Why Aren't Floods?. *Water Resources Research*, 54(11), pp.8545-8551.
- Pendergrass, A.G., 2018. What precipitation is extreme?. *Science*, 360(6393), pp.1072-1073.
- Rasmussen, K.L., Prein, A.F., Rasmussen, R.M., Ikeda, K. and Liu, C., 2017. Changes in the convective population and thermodynamic environments in convection-permitting regional climate simulations over the United States. *Climate Dynamics*, pp.1-26.
- Maidment, R.I., Allan, R.P. and Black, E., 2015. Recent observed and simulated changes in precipitation over Africa. *Geophysical Research Letters*, 42(19), pp.8155-8164.
- Taylor, R.G., Todd, M.C., Kongola, L., Maurice, L., Nahozya, E., Sanga, H. and MacDonald, A.M., 2013. Evidence of the dependence of groundwater resources on extreme rainfall in East Africa. *Nature Climate Change*, 3(4), p.374.
- Taylor, C.M., Belušić, D., Guichard, F., Parker, D.J., Vischel, T., Bock, O., Harris, P.P., Janicot, S., Klein, C. and Panthou, G., 2017. Frequency of extreme Sahelian storms tripled since 1982 in satellite observations. *Nature*, 544(7651), p.475.

Reviewer #2

This is my second reading of this manuscript, as I had already reviewed the version of this paper submitted to *Nature Climate Change*. At that time, I had appreciated the work, recognizing that convective scale simulations are really needed to better understand present climate characteristics and its future evolution especially for regions such as tropical Africa. My main criticism, i.e. the too short duration of the simulation, has somehow been addressed, as, now, 10 year simulations have been used. Although this duration is still relatively short to fully grasp all the natural variability and associated uncertainties (both in the present and in the future), I do acknowledge that long term (centennial transient) convection-permitting continental-scale simulation will not be available for a while. So, to this regard, this work is indeed innovative and important. I do also appreciate the effort of the authors to address my other points regarding clarifying part of the text and methodology and to revise the supporting literature, that is more complete now.

We thank the reviewer for their positive comment, and are glad that our earlier revisions have addressed many of their previous concerns.

The emphasis of dry spell duration in this new version is also interesting; although the fact that increasing model resolution (and better representation of physical process) will increase the occurrence and intensity of extreme events is not really completely new, I found interesting for instance, from Supplementary Figs 3 and 4 that CP4 projects, compared to R25 a much shorter but more intense wet season over most of tropical Africa, (which again confirms previous GCMs and RCMs based studies) that could be possibly emphasized more for its obvious potential impact on society and economy of Africa.

In this manuscript we do not look at the length of the wet season, but instead define this to be the 3-month period with the highest mean rainfall. Supplementary Figs. 3 and 4 have been moved to the main text and are now Figs. 3 and 4. These figures show the occurrence and intensity of 3hrly rainfall, for this 3-month period. It is true that over most of tropical Africa CP4 produces a wet season in which rainfall is less frequent but more intense than in R25. This is similar to Saeed et al 2013, which showed the more intense and intermittent nature of rainfall at higher resolution.

We have added a statement to the paper emphasizing this result, with reference to Saeed et al 2013:

“In CP4A, wet season rainfall is less frequent but more intense over most of tropical Africa, confirming previous RCM studies that showed a similar result on increasing resolution (Saeed et al, 2013).”